# Antisense oligonucleotide therapy rescues disturbed brain rhythms and sleep in juvenile and adult mouse models of Angelman syndrome

**Dongwon Lee[1,2†], Wu Chen[1,2*†], Heet Naresh Kaku[1,2†], Xinming Zhuo[3†‡], Eugene S Chao[1,2], Armand Soriano[4], Allen Kuncheria[1], Stephanie Flores[1], Joo Hyun Kim[1,2], Armando Rivera[1,2], Frank Rigo[4], Paymaan Jafar-nejad[4], Arthur L Beaudet[3§], Matthew S Caudill[1,5], Mingshan Xue[1,2,3*]**

[1]Department of Neuroscience, Baylor College of Medicine, Houston, United States; [2]The Cain Foundation Laboratories, Jan and Dan Duncan Neurological Research Institute at Texas Children's Hospital, Houston, United States; [3]Department of Molecular and Human Genetics, Baylor College of Medicine, Houston, United States; [4]Ionis Pharmaceuticals, Carlsbad, United States; [5]Jan and Dan Duncan Neurological Research Institute at Texas Children's Hospital, Houston, United States

**\*For correspondence:**
wu.chen@bcm.edu (WC);
mxue@bcm.edu (MX)

[†]These authors contributed equally to this work

**Present address:** [‡]CLIA Laboratory, The Jackson Laboratory for Genomic Medicine, Farmington, United States; [§]Luna Genetics, Houston, United States

**Abstract** UBE3A encodes ubiquitin protein ligase E3A, and in neurons its expression from the paternal allele is repressed by the UBE3A antisense transcript (UBE3A-ATS). This leaves neurons susceptible to loss-of-function of maternal UBE3A. Indeed, Angelman syndrome, a severe neurodevelopmental disorder, is caused by maternal UBE3A deficiency. A promising therapeutic approach to treating Angelman syndrome is to reactivate the intact paternal UBE3A by suppressing UBE3A-ATS. Prior studies show that many neurological phenotypes of maternal Ube3a knockout mice can only be rescued by reinstating Ube3a expression in early development, indicating a restricted therapeutic window for Angelman syndrome. Here, we report that reducing Ube3a-ATS by antisense oligonucleotides in juvenile or adult maternal Ube3a knockout mice rescues the abnormal electroencephalogram (EEG) rhythms and sleep disturbance, two prominent clinical features of Angelman syndrome. Importantly, the degree of phenotypic improvement correlates with the increase of Ube3a protein levels. These results indicate that the therapeutic window of genetic therapies for Angelman syndrome is broader than previously thought, and EEG power spectrum and sleep architecture should be used to evaluate the clinical efficacy of therapies.

## Editor's evaluation

We believe this study contributes importantly to the literature and in particular, provides support for the potential value of further postnatal rescue experiments in animal models and perhaps future trials in patients.

## Introduction

Angelman syndrome is a rare neurodevelopmental disorder characterized by severe intellectual disability, developmental delay, speech impairment, motor dysfunction, behavioral uniqueness, microcephaly, sleep disturbance, seizures, and abnormal electroencephalogram (EEG) (*Williams et al., 2006*; *Bird, 2014*; *Buiting et al., 2016*). It is caused by loss-of-function of the maternally derived

*UBE3A* gene encoding ubiquitin protein ligase E3A (*Kishino et al., 1997*; *Matsuura et al., 1997*). *UBE3A* is expressed from both maternal and paternal alleles in non-neuronal cells, but is paternally imprinted in neurons (*Albrecht et al., 1997*; *Rougeulle et al., 1997*; *Judson et al., 2014*). Imprinted expression of *UBE3A* or silence of the paternal allele in neurons is due to a long non-coding RNA, *UBE3A* antisense transcript (*UBE3A-ATS*). *UBE3A-ATS* is expressed from the paternally inherited chromosome and localized in the nucleus to repress *UBE3A in cis* through a transcriptional collision mechanism (*Meng et al., 2012*; *Meng et al., 2013*). Thus, loss-of-function of the maternal *UBE3A* leads to the absence of functional UBE3A proteins in neurons. Maternal *UBE3A* deficiency can result from deletions in the chromosomal region 15q11–q13 containing *UBE3A*, which account for about 70% of Angelman syndrome patients, mutations in *UBE3A* gene, paternal uniparental disomy of chromosome 15, or an imprinting defect (*Williams et al., 2006*; *Bird, 2014*; *Buiting et al., 2016*).

Studies in rodent models carrying maternal *Ube3a* loss-of-function mutations have provided insights into Angelman syndrome mechanisms and identified therapeutic strategies. Many disease relevant phenotypes were reported in these Angelman syndrome models (*Margolis et al., 2015*; *Rotaru et al., 2020*; *Yang, 2020*), and some of them are robust and reproducible in different models and laboratories, including motor impairments (e.g., poor performance in rotarod and open-field tests), decreased innate marble burying and nest building behaviors, cortical hyperexcitability (e.g, poly-spikes in EEG), altered EEG power spectrum and sleep pattern, increased susceptibility to seizure induction, and reduced synaptic long-term potentiation (*Jiang et al., 1998*; *Miura et al., 2002*; *Weeber et al., 2003*; *Colas et al., 2005*; *Yashiro et al., 2009*; *Jiang et al., 2010*; *Huang et al., 2013*; *Meng et al., 2013*; *Ehlen et al., 2015*; *Silva-Santos et al., 2015*; *Judson et al., 2016*; *Born et al., 2017*; *Sidorov et al., 2017*; *Sonzogni et al., 2018*; *Gu et al., 2019*; *Copping and Silverman, 2021*). Thus, these phenotypes are suitable for evaluating the effects of potential disease-modifying therapies even though some of them may not be clinically relevant. By genetically reinstating *Ube3a* expression from the maternal allele at different developmental ages, rescue experiments in mice show that most of these neurological functions require *Ube3a* during late embryonic and early postnatal development (*Silva-Santos et al., 2015*; *Rotaru et al., 2018*; *Gu et al., 2019*; *Sonzogni et al., 2020*), suggesting that the therapeutic window for Angelman syndrome may be limited to very young ages.

Two categories of therapeutic strategies are being actively pursued for Angelman syndrome. One is to target the downstream substrates of UBE3A protein, and the other is to restore *UBE3A* gene expression (*Margolis et al., 2015*; *Yang, 2020*; *Copping et al., 2021*; *Elgersma and Sonzogni, 2021*; *Markati et al., 2021*). Since the paternal *UBE3A* allele is intact in Angelman syndrome, an attractive approach is to reactivate the silenced paternal *UBE3A* by suppressing *UBE3A-ATS* expression (*Figure 1A*). Indeed, reducing *Ube3a-ATS* levels in mice through genetic manipulations or topoisomerase inhibition results in unsilencing of the paternal *Ube3a* (*Huang et al., 2011*; *Meng et al., 2012*; *Meng et al., 2013*). Antisense oligonucleotides (ASOs) or CRISPR/Cas9 targeting the mouse *Ube3a-ATS* can also upregulate paternal *Ube3a* expression. Administering ASO or adeno-associated virus (AAV) expressing CRISPR/Cas9 to newborn, but not older, maternal *Ube3a* knockout mice rescues a subset of phenotypes (*Meng et al., 2015*; *Wolter et al., 2020*; *Milazzo et al., 2021*; *Schmid et al., 2021*). Currently, three Phase 1 or Phase 1/2 clinical trials with ASOs targeting *UBE3A-ATS* are underway. Another approach is to directly express an exogenous copy of *UBE3A* by AAV, which also only rescues a subset of phenotypes of maternal *Ube3a* knockout mice when administered postnatally (*Daily et al., 2011*; *Judson et al., 2021*). The reversibility of neurological phenotypes at different ages is summarized in *Supplementary file 1*. These results imply that gene-targeted therapies may have to be administered to late trimester fetuses or newborns to treat Angelman syndrome, as the first postnatal week in mice is generally assumed to be equivalent to the third trimester of human gestation with regard to the central nervous system development (*Zeiss, 2021*). This timing of intervention would be difficult because currently Angelman syndrome is diagnosed after at least the first 6 months of life and typically between 1 and 4 years of age (*Williams et al., 2006*).

However, the reversibility of three robust and clinically relevant phenotypes of Angelman mouse models remains untested, namely cortical hyperexcitability (e.g, poly-spikes), altered EEG power spectrum, and sleep disturbance. Alterations in EEG power spectrum, particularly an increase in the power of low frequency oscillations (i.e., 1–30 Hz), are well documented in both Angelman syndrome patients and rodent models (*Born et al., 2017*; *Sidorov et al., 2017*; *Frohlich et al., 2019*; *Born et al., 2021*; *Copping and Silverman, 2021*). This brain rhythm is a potential biomarker for assessing

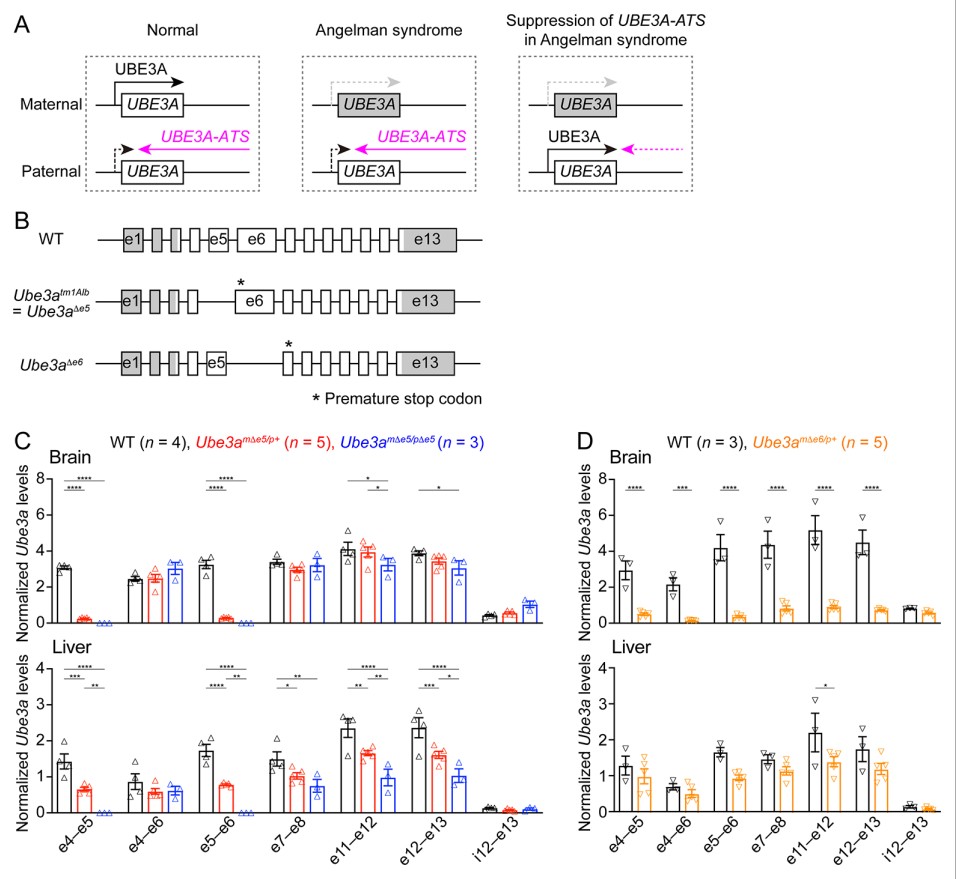

**Figure 1.** *Ube3a* mRNA is diminished in the brain of a new maternal *Ube3a* knockout mouse but remains in a previously generated Angelman syndrome mouse model. (**A**) Schematics of *UBE3A* imprinting and Angelman syndrome. Left, in normal neurons, UBE3A proteins are only produced from the maternal copy of *UBE3A* because the paternal copy is silenced by *UBE3A-ATS*. Middle, deficiency of the maternal *UBE3A* (gray) leads to the loss of UBE3A proteins in neurons and causes Angelman syndrome. Right, suppressing *UBE3A-ATS* expression leads to the unsilencing of the paternal *UBE3A*. (**B**) Genomic structures of *Ube3a* wild-type (WT), *Δe5* (also known as *tm1Alb*), and *Δe6* alleles. The boxes indicate exons (e) 1–13. The white and gray regions indicate the coding and non-coding exon sequences of the longest *Ube3a* transcript, respectively. In the *Δe5* and *Δe6* alleles, exons 5 and 6 are deleted, resulting in a premature stop codon in exons 6 and 7, respectively. (**C**) *Ube3a* transcript levels were measured from the brains and livers of WT, *Ube3a*$^{mΔe5/p+}$ and *Ube3a*$^{mΔe5/pΔe5}$ mice using primer sets targeting different exons or introns as indicated in the figure. *Ube3a* mRNA levels were normalized by the *Gapdh* mRNA levels. Except the deleted exon 5, other exons in the brains of *Ube3a*$^{mΔe5/p+}$ and *Ube3a*$^{mΔe5/pΔe5}$ mice remain at the similar levels as WT mice. (**D**) Similar to (**C**), but for WT and *Ube3a*$^{mΔe6/p+}$ mice. *Ube3a* transcript is greatly reduced in the *Ube3a*$^{mΔe6/p+}$ mouse brains. The numbers of tested mice are indicated in the figure. Each symbol represents one mouse. Bar graphs are mean ± standard error of the mean (SEM). Two-way analysis of variance (ANOVA) with Tukey (**C**) or Šídák (**D**) multiple comparison test for all pairs of groups, $*p < 0.05$, $**p < 0.01$, $***p < 0.001$, $****p < 0.0001$.

clinical symptoms because the power in the delta range (2–4 Hz) correlates with symptom severity in Angelman syndrome patients (*Hipp et al., 2021*; *Ostrowski et al., 2021*). Similarly, sleep disturbance, another prominent feature of Angelman syndrome (*Spruyt et al., 2018*), is also recapitulated in the mouse models (*Colas et al., 2005*; *Ehlen et al., 2015*; *Copping and Silverman, 2021*). Hence, to support the ongoing and planned clinical trials, it is crucial to determine to what extent the EEG and sleep deficits are reversible in juvenile and adult Angelman syndrome mouse models because these developmental ages are more clinically relevant than the neonatal period. To address this question, we generated a new mouse *Ube3a* null allele and administered a single dose of ASOs targeting *Ube3a-ATS* to juvenile or adult maternal *Ube3a* knockout mice. We first systematically determined the levels of *Ube3a-ATS* and *Ube3a* transcripts and Ube3a proteins across different brain regions at

different timepoints post ASO administration and then evaluated the corresponding EEG and sleep phenotypes.

## Results

### Generation of a new *Ube3a* null allele in mice

Our goal was to assess the effect of *Ube3a-ATS*-targeted ASOs in a mouse model of Angelman syndrome. The widely used mouse model is a *Ube3a* knockout allele (*Ube3a^{tm1Alb}*, referred to as *Ube3a^{Δe5}* here to be distinguished from the new allele) that deletes exon 5 (previously named as exon 2), resulting in a premature stop codon in exon 6 (*Jiang et al., 1998*; *Figure 1B*). We performed reverse transcription droplet digital PCR (RT-ddPCR) analyses on this *Ube3a* knockout allele with primer sets targeting different exons. Exons 4, 6, and other exons downstream of the deleted exon 5 were still transcribed in the brains of adult heterozygous maternal (*Ube3a^{mΔe5/p+}*) and homozygous (*Ube3a^{mΔe5/pΔe5}*) mutant mice at a level comparable to wild-type (WT) mice (*Figure 1C*), possibly due to an escape from nonsense-mediated mRNA decay or an alternative start site. Similarly, *Ube3a* mRNA was only modestly reduced in the livers of *Ube3a^{mΔe5/p+}* and *Ube3a^{mΔe5/pΔe5}* mice (*Figure 1C*). Although this knockout allele produces very little full-length functional Ube3a proteins in the brain (*Judson et al., 2014*; *Grier et al., 2015*), we sought to create a new *Ube3a* null allele with diminished *Ube3a* mRNA to facilitate the evaluation of ASO efficacy at the transcript level. CRISPR/Cas9 was used to delete the largest *Ube3a* coding exon, exon 6. The resulting allele (*Ube3a^{Δe6}*) carries a premature stop codon in exon 7 (*Figure 1B*). RT-ddPCR analyses of adult heterozygous maternal mutant mice (*Ube3a^{mΔe6/p+}*) showed that *Ube3a* mRNA was diminished in the brain and reduced in the liver as compared to WT mice (*Figure 1D*). Western blots revealed that Ube3a protein levels in different brain regions of *Ube3a^{mΔe6/p+}* mice were 2–17% of those in WT mice when they were at 6 weeks of age or older (see below). Thus, both *Ube3a* mRNA and proteins are diminished in the *Ube3a^{mΔe6/p+}* mouse brains. Furthermore, *Ube3a^{mΔe6/p+}* mice showed similar rotarod and marble burying phenotypes to previously reported deficits in *Ube3a^{mΔe5/p+}* mice (*Shi et al., 2022*).

### ASOs targeting *Ube3a-ATS* non-coding RNA upregulate paternal *Ube3a-YFP* expression

To increase the paternal expression of *Ube3a* in mice, we used two mouse-specific ASOs to downregulate the *Ube3a-ATS* levels. The first one (Ube3a-as) is complementary to a region downstream of the *Snord115* small nuclear RNA cluster, and this sequence was also targeted by the 'ASO B' used in a previous study (*Meng et al., 2015*). The second one (Snord115) is complementary to a sequence that repeats 110 times in the *Snord115* RNAs. We first tested these two ASOs in 8-week-old WT mice by administering them via intracerebroventricular (ICV) injection. Both Ube3a-as ASO and Snord115 ASO decreased *Ube3a-ATS* transcripts and increased *Ube3a* mRNA levels (*Figure 2—figure supplement 1A, B*). To further test the effect on paternal expression of *Ube3a* and visualize the distribution of these two ASOs in the brain, we used paternal *Ube3a^{YFP}* mice (*Ube3a^{m+/pYFP}*) carrying a *yellow fluorescent protein* (YFP)-tagged *Ube3a* (*Dindot et al., 2008*), as downregulating *Ube3a-ATS* is expected to reactivate the paternal *Ube3a-YFP* allele. A non-targeting control ASO, Ube3a-as ASO, or Snord115 ASO was administered to the brains of 3-month-old *Ube3a^{m+/pYFP}* mice by a single unilateral ICV injection. We visualized Ube3a-YFP expression by immunostaining of YFP 18 days post ASO injection. Maternal *Ube3a^{YFP}* mice (*Ube3a^{mYFP/p+}*) exhibited strong Ube3a-YFP in the brain, whereas *Ube3a^{m+/pYFP}* mice receiving the control ASO showed little expression (*Figure 2A, B*). Ube3a-as ASO and Snord115 ASO increased Ube3a-YFP expression in many brain regions of *Ube3a^{m+/pYFP}* mice to 40–90% and 20–60% of that of *Ube3a^{mYFP/p+}* mice, respectively (*Figure 2A, B*; *Figure 2—figure supplement 1C*). Although we did not specifically examine different cell types, YFP was observed in several types of GABAergic neurons including cerebellar Purkinje cells, olfactory bulb granule cells, striatal neurons, and interneurons in cortical layer 1, hippocampal stratum oriens, and cerebellar molecular layer (*Figure 2B*). Thus, both Ube3a-as ASO and Snord115 ASO can broadly reactivate the paternal *Ube3a-YFP* allele in neurons including GABAergic neurons throughout the mouse brain *in vivo*.

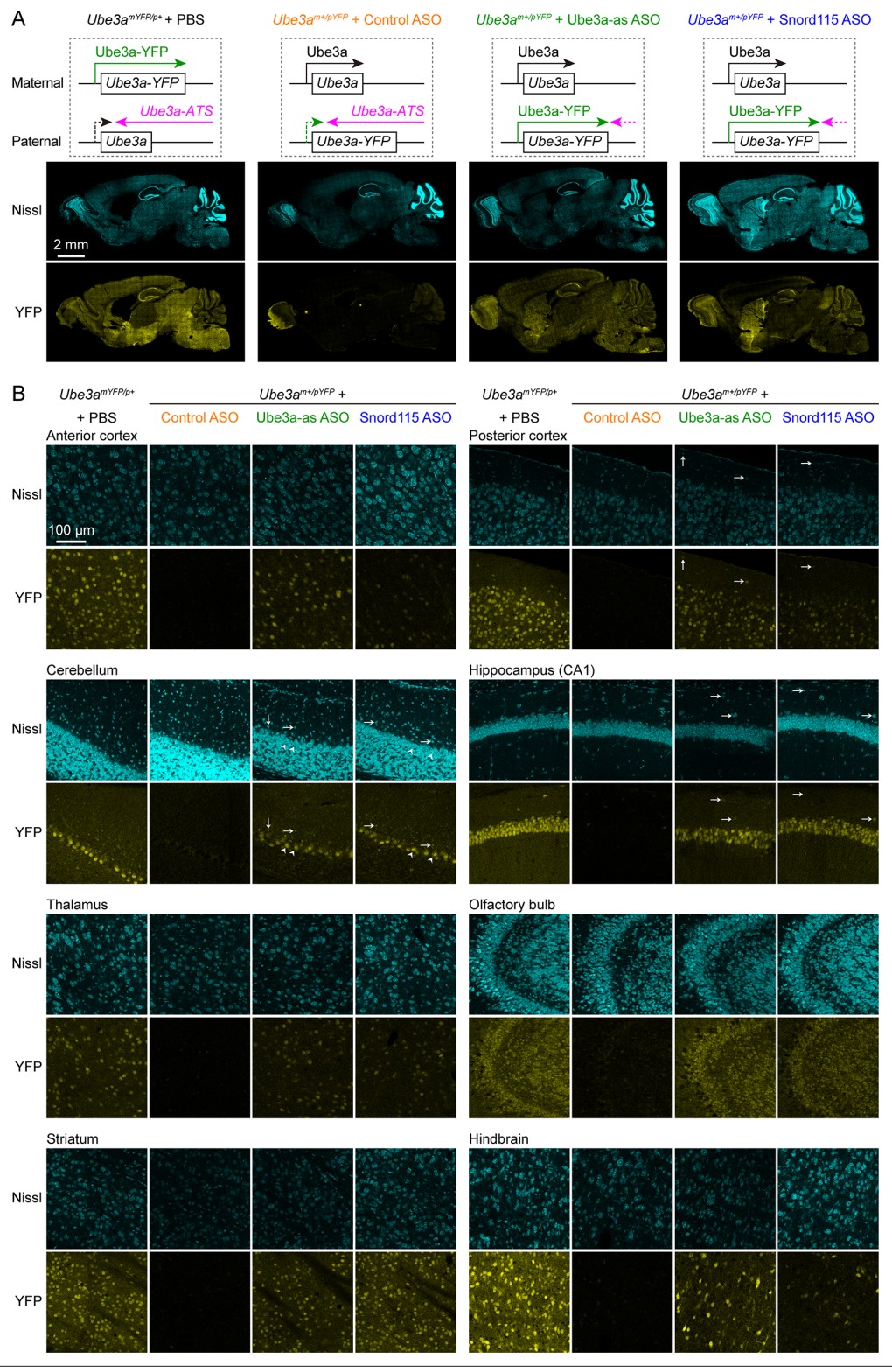

**Figure 2.** Antisense oligonucleotides (ASOs) targeting *Ube3a-ATS* reactivate paternal *Ube3a-YFP* expression. (**A**) Schematics of Ube3a-YFP expression (upper panels) and representative fluorescent images of sagittal brain sections (lower panels) from maternal *Ube3a^YFP* mice (*Ube3a^{mYFP/p+}*, *n* = 2) injected with phosphate-buffered saline (PBS) and paternal *Ube3a^YFP* mice (*Ube3a^{m+/pYFP}*) injected with control (*n* = 2), Ube3a-as (*n* = 2), or Snord115 (*n* = 2) ASO. Sections were stained with fluorescent Nissl and an antibody recognizing YFP. Ube3a-YFP proteins are

*Figure 2 continued on next page*

*Figure 2 continued*

produced from the maternal copy of *Ube3a-YFP* in *Ube3a^{mYFP/p+}* mice, but not from the paternal copy in *Ube3a^{m+/pYFP}* mice injected with control ASO because the paternal copy is silenced by *Ube3a-ATS*. Both Ube3a-as and Sord115 ASOs can suppress *Ube3a-ATS* expression in *Ube3a^{m+/pYFP}* mice and broadly reactivate the paternal *Ube3a-YFP* expression in the brains. (**B**) Similar to (**A**), but for images of eight different brain regions at high magnification. Arrows indicate YFP-positive GABAergic interneurons in cortical layer 1, cerebellar molecular layer, and hippocampal stratum oriens. Arrow heads indicate YFP-positive cerebellar Purkinje cells.

The online version of this article includes the following figure supplement(s) for figure 2:

**Figure supplement 1.** Antisense oligonucleotides (ASOs) targeting *Ube3a-ATS* increase *Ube3a* transcripts in wild-type (WT) mice and *Ube3a-YFP* expression in *Ube3a^{m+/pYFP}* mice.

## Ube3a-as ASO and Snord115 ASO reactivate paternal *Ube3a* expression in *Ube3a^{mΔe6/p+}* mice

We sought to assess the efficacy of Ube3a-as ASO and Snord115 ASO in our new mouse model of Angelman syndrome across brain regions and over time by systematically determining the total Ube3a protein, *Ube3a* mRNA, and *Ube3a-ATS* levels. We first injected male and female *Ube3a^{mΔe6/p+}* mice with control, Ube3a-as, or Snord115 ASO and their sex- and age-matched WT littermates with control ASO in parallel at the juvenile age (postnatal days 21.9 ± 0.1 [mean ± standard error of the mean, SEM], range 21–24, *n* = 91). Brain tissues were harvested from eight different regions of both hemispheres at 3, 6, and 10 weeks post ASO injection (*Figure 3A*). We used the hemispheres ipsilateral to the ASO injection site for Western blot analyses and the corresponding contralateral hemispheres for reverse transcription quantitative real-time PCR (RT-qPCR) analyses. At 3 weeks post ASO injection, Ube3a-as ASO and Snord115 ASO increased Ube3a protein levels in *Ube3a^{mΔe6/p+}* mice to about 28–71% of the WT levels in different brain regions (*Figure 3B, D, E*; *Figure 3—figure supplement 1A*; *Figure 3—figure supplement 2*). Correlated with this result, *Ube3a-ATS* levels were downregulated by 29–73% (*Figure 4A, B*, *Figure 4—figure supplement 1*) and *Ube3a* mRNA levels were increased to about 22–57% of the WT levels (*Figure 4A, B*, *Figure 4—figure supplement 1*). The effects of Ube3a-as ASO remained stable for at least 10 weeks. However, the Ube3a protein and *Ube3a* mRNA levels in Snord115 ASO-treated *Ube3a^{mΔe6/p+}* mice markedly decreased at 6 weeks post injection and reached to the levels of control ASO-treated *Ube3a^{mΔe6/p+}* mice in most brain regions by 10 weeks post injection (*Figure 3B, D, E*; *Figure 3—figure supplement 1A*; *Figure 3—figure supplement 2*; *Figure 4A, B*; *Figure 4—figure supplement 1*). Correspondingly, by 10 weeks post injection the *Ube3a-ATS* levels in the Snord115 ASO-treated *Ube3a^{mΔe6/p+}* mice returned to those in control ASO-treated WT or *Ube3a^{mΔe6/p+}* mice (*Figure 4A, B*; *Figure 4—figure supplement 1*).

We further tested the ASOs in adult mice (postnatal days 64.6 ± 0.5 [mean ± SEM], range 56–72, *n* = 83). At 3 weeks post ASO injection, Ube3a-as ASO and Snord115 ASO also significantly reduced *Ube3a-ATS* levels and increased *Ube3a* mRNA and Ube3a protein levels in *Ube3a^{mΔe6/p+}* mice (*Figure 3C, D, F*; *Figure 3—figure supplement 1B*; *Figure 3—figure supplement 2*; *Figure 4A, B*; *Figure 4—figure supplement 1*). However, the upregulation of Ube3a proteins appeared to be slightly less effective in some brain regions (e.g., posterior cortex, thalamus and hypothalamus, and midbrain and hindbrain) than injecting ASOs to juvenile mice (*Figure 3E, F*). Furthermore, the effects of Ube3a-as ASO modestly decreased over the course of 10 weeks post injection, and the reduction in efficacies over time was much more evident for Snord115 ASO (*Figure 3F*; *Figure 4A, B*; *Figure 4—figure supplement 1*).

*Ube3a* gene in mice produces two Ube3a protein isoforms that differ by 21 amino acids at the N terminus. The short Ube3a isoform 3 is the dominant isoform and mainly localized in the nucleus, whereas the long isoform 2 is expressed at 20–30% of the isoform 3 level and primarily distributed in the cytosol (*Miao et al., 2013*; *Avagliano Trezza et al., 2019*). Mice selectively lacking the Ube3a isoform 3 recapitulate several key phenotypes of maternal *Ube3a* knockout mice, showing the critical role of the nuclear Ube3a isoform 3 in Angelman syndrome (*Avagliano Trezza et al., 2019*). Thus, we performed additional Western blot analyses and found that both Ube3a isoforms were similarly upregulated by Ube3a-as ASO or Snord115 ASO in different brain regions of *Ube3a^{mΔe6/p+}* mice (*Figure 3—figure supplement 3*; *Figure 3—figure supplement 4*), confirming the successful reactivation of the critical isoform 3.

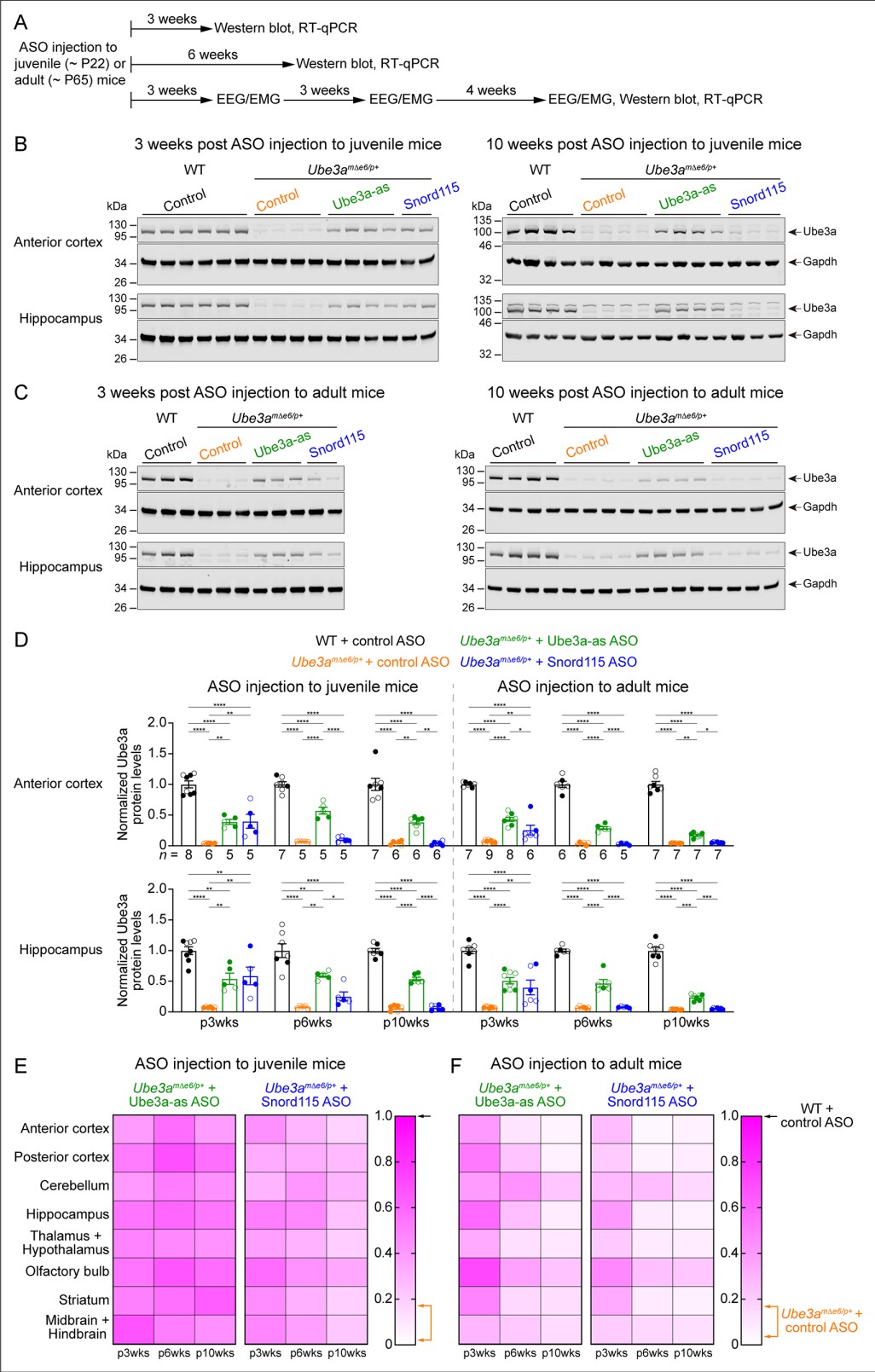

**Figure 3.** Antisense oligonucleotides (ASOs) targeting *Ube3a-ATS* upregulate Ube3a protein in *Ube3a^{mΔe6/p+}* mice. (**A**) Experimental designs and timelines. ASOs were injected into three cohorts of juvenile mice around postnatal day 22 (P22). Protein and RNA were measured from two cohorts of mice at 3 and 6 weeks post ASO injection. Electroencephalogram (EEG) and electromyogram (EMG) were measured from the third cohort of mice at 3, 6, and

*Figure 3 continued on next page*

*Figure 3 continued*

10 weeks post ASO injection, and protein and RNA were measured after the last EEG/EMG recording. The same experiments were performed for three cohorts of adult mice injected with ASOs around P65. (**B, C**) ASOs were injected into juvenile (**B**) or adult (**C**) mice. Representative Western blots at 3 and 10 weeks post ASO injection from the anterior cortex and hippocampus of wild-type (WT) mice injected with control ASO and $Ube3a^{m\Delta e6/p+}$ mice with control, Ube3a-as, or Snord115 ASO. Gapdh, a housekeeping protein as loading control. (**D**) Summary data of normalized Ube3a protein levels from the anterior cortex (upper panel) and hippocampus (lower panel) at 3, 6, and 10 weeks post ASO injection indicated by p3wks, p6wks, and p10wks, respectively. Ube3a levels were first normalized by the Gapdh levels and then by the average Ube3a levels of all WT mice from the same blot. Ube3a levels are diminished in control ASO-treated $Ube3a^{m\Delta e6/p+}$ mice as compared to control ASO-treated WT mice. The upregulation of Ube3a protein by Ube3a-as ASO is evident up to 10 weeks post ASO injection, whereas the effect of Snord115 ASO diminishes over time. The numbers of tested mice are indicated in the figure. Each filled (male) or open (female) circle represents one mouse. Bar graphs are mean ± standard error of the mean (SEM). One-way analysis of variance (ANOVA) with Tukey multiple comparison test for all pairs of groups, *$p < 0.05$, **$p < 0.01$, ***$p < 0.001$, ****$p < 0.0001$. (**E, F**) Heat maps showing the normalized Ube3a protein levels from different brain regions of Ube3a-as or Snord115 ASO-treated juvenile (**E**) and adult (**F**) $Ube3a^{m\Delta e6/p+}$ mice at 3, 6, and 10 weeks post ASO injection. In the color scales, 1 represents the Ube3a levels in control ASO-treated WT mice for each brain region (black arrows), and the orange arrows indicate the range of Ube3a levels in control ASO-treated $Ube3a^{m\Delta e6/p+}$ mice.

The online version of this article includes the following source data and figure supplement(s) for figure 3:

**Source data 1.** Raw images of the Western blots in *Figure 3B*.

**Source data 2.** Raw images of the Western blots in *Figure 3C*.

**Figure supplement 1.** Antisense oligonucleotides (ASOs) targeting *Ube3a-ATS* upregulate Ube3a protein in different brain regions of $Ube3a^{m\Delta e6/p+}$ mice (Part I).

**Figure supplement 1—source data 1.** Raw images of the Western blots in *Figure 3—figure supplement 1A*, left panels.

**Figure supplement 1—source data 2.** Raw images of the Western blots in *Figure 3—figure supplement 1A*, right panels.

**Figure supplement 1—source data 3.** Raw images of the Western blots in *Figure 3—figure supplement 1B*, left panels.

**Figure supplement 1—source data 4.** Raw images of the Western blots in *Figure 3—figure supplement 1B*, right panels.

**Figure supplement 2.** Antisense oligonucleotides (ASOs) targeting *Ube3a-ATS* upregulate Ube3a protein in different brain regions of $Ube3a^{m\Delta e6/p+}$ mice (Part II).

**Figure supplement 3.** Ube3a isoforms 2 and 3 are similarly upregulated by *Ube3a-ATS*-targeted antisense oligonucleotides (ASOs) in juvenile $Ube3a^{m\Delta e6/p+}$ mice.

**Figure supplement 3—source data 1.** Raw images of the Western blots in *Figure 3—figure supplement 3A*, upper panels.

**Figure supplement 3—source data 2.** Raw images of the Western blots in *Figure 3—figure supplement 3A*, lower panels.

**Figure supplement 4.** Ube3a isoforms 2 and 3 are similarly upregulated by *Ube3a-ATS*-targeted antisense oligonucleotides (ASOs) in adult $Ube3a^{m\Delta e6/p+}$ mice.

**Figure supplement 4—source data 1.** Raw images of the Western blots in *Figure 3—figure supplement 4A*, upper panels.

**Figure supplement 4—source data 2.** Raw images of the Western blots in *Figure 3—figure supplement 4A*, lower panels.

**Figure supplement 5.** Similar increase of Ube3a protein by two different doses of antisense oligonucleotides (ASOs) in $Ube3a^{m\Delta e6/p+}$ mice.

Finally, we examined the relationships among total *Ube3a-ATS*, *Ube3a* mRNA, and Ube3a protein levels across individual mice and timepoints. *Ube3a-ATS* levels negatively correlated with Ube3a protein and *Ube3a* mRNA levels in $Ube3a^{m\Delta e6/p+}$ mice (*Figure 4C*; *Figure 4—figure supplement 2A*), and Ube3a protein levels positively correlated with *Ube3a* mRNA levels (*Figure 4—figure supplement 2B*). Since the protein and transcript levels were measured at the same post ASO injection timepoints from the ipsilateral and contralateral hemispheres, respectively, these correlations indicate a broad distribution of ASOs in the mouse brains and a spatiotemporally comparable pattern between

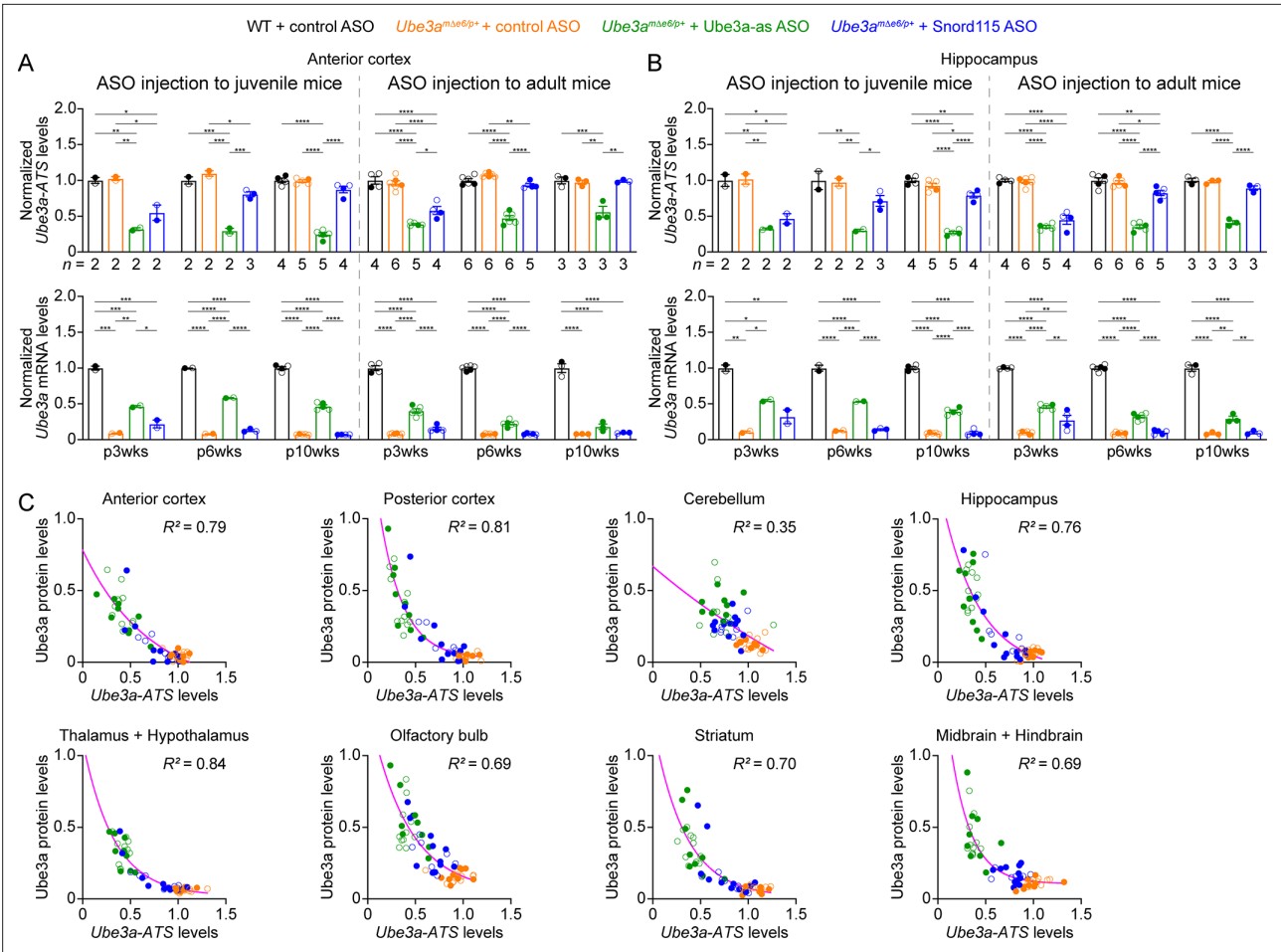

**Figure 4.** Antisense oligonucleotides (ASOs) targeting *Ube3a-ATS* reduce *Ube3a-ATS* and increase *Ube3a* transcripts in *Ube3a^{mΔe6/p+}* mice. (**A, B**) Juvenile or adult wild-type (WT) mice were injected with control ASO and *Ube3a^{mΔe6/p+}* mice with control, Ube3a-as, or Snord115 ASO. Summary data show the normalized *Ube3a-ATS* (upper panels) and *Ube3a* (lower panels) transcript levels from the anterior cortex (**A**) and hippocampus (**B**) at 3, 6, and 10 weeks post ASO injection indicated by p3wks, p6wks, and p10wks, respectively. *Ube3a-ATS* and *Ube3a* mRNA levels were first normalized by the *Gapdh* levels and then by the average *Ube3a-ATS* and *Ube3a* mRNA levels of all WT mice, respectively. The downregulation of *Ube3a-ATS* and upregulation of *Ube3a* mRNA by Ube3a-as ASO are evident up to 10 weeks post ASO injection, whereas the effect of Snord115 ASO diminishes over time. The numbers of tested mice are indicated in the figure. Bar graphs are mean ± standard error of the mean (SEM). One-way analysis of variance (ANOVA) with Tukey multiple comparison test for all pairs of groups, *$p < 0.05$, **$p < 0.01$, ***$p < 0.001$, ****$p < 0.0001$. (**C**) The negative correlations between *Ube3a-ATS* transcript levels and Ube3a protein levels from different brain regions of *Ube3a^{mΔe6/p+}* mice injected with control, Ube3a-as, or Snord115 ASO were fitted with a one phase exponential decay ($Y = ae^{-kX} + b$; *X*, *Ube3a-ATS* transcript levels; *Y*, Ube3a protein levels; *a*, *b*, *k*, constants). Data from 3, 6, and 10 weeks post ASO injection into juvenile and adult mice were all included. Each filled (male) or open (female) circle represents one mouse. $R^2$ indicates the goodness of fit.

The online version of this article includes the following figure supplement(s) for figure 4:

**Figure supplement 1.** Antisense oligonucleotides (ASOs) targeting *Ube3a-ATS* reduce *Ube3a-ATS* and increase *Ube3a* transcripts in different brain regions of *Ube3a^{mΔe6/p+}* mice (Part I).

**Figure supplement 2.** Antisense oligonucleotides (ASOs) targeting *Ube3a-ATS* reduce *Ube3a-ATS* and increase *Ube3a* transcripts in different brain regions of *Ube3a^{mΔe6/p+}* mice (Part II).

the changes in transcripts and proteins. Taken together, our results demonstrate that a single unilateral ICV injection of ASO targeting *Ube3a-ATS* in *Ube3a^{mΔe6/p+}* mice leads to a long-lasting downregulation of this transcript and reactivation of the paternal *Ube3a* allele throughout the brains, and the upregulation of Ube3a proteins by Ube3a-as ASO can last at least 10 weeks.

## Reactivation of paternal *Ube3a* expression alleviates abnormal EEG rhythmic activity in *Ube3a$^{m\Delta e6/p+}$* mice

Maternal *Ube3a* deficiency in mice causes altered brain rhythms, sleep disturbance, and epileptiform activity (e.g., cortical poly-spikes), all of which can be examined by chronic video-EEG and electromyogram (EMG) recordings. Thus, to determine if upregulation of paternal *Ube3a* expression can reverse these phenotypes in *Ube3a$^{m\Delta e6/p+}$* mice, we injected male and female *Ube3a$^{m\Delta e6/p+}$* mice with control, Ube3a-as, or Snord115 ASO and their sex- and age-matched WT littermates with control ASO in parallel at the juvenile (postnatal days 21.5 ± 0.1 [mean ± SEM], range 21–24, $n$ = 35) or adult (postnatal days 62.5 ± 0.6 [mean ± SEM], range 56–66, $n$ = 28) age. Intracranial EEG from the frontal, somatosensory, and visual cortices and EMG from the neck muscles of each mouse were recorded at 3, 6, and 10 weeks post ASO injection (*Figure 3A*, *Figure 5A*). To avoid bias, we evenly sampled 6 out of 24 hr of the EEG/EMG data for power spectrum and poly-spikes analyses and used 24 hr of data for sleep scoring (see Materials and methods).

We first removed artifacts and then computed the absolute power spectral densities (PSDs) of EEG signals including all brain states (*Figure 5—figure supplement 1*). To control for the variations caused by different impedances across electrodes and mice, we normalized PSDs by the total power within 1–100 Hz to obtain the relative PSDs. The relative PSDs from the frontal cortex of control ASO-treated *Ube3a$^{m\Delta e6/p+}$* mice were higher at 4–25 Hz and lower at 40–100 Hz than those of control ASO-treated WT mice (*Figure 5B–D, F–H*; *Figure 5—figure supplement 2*). Thus, we further computed the relative power in the frequency bands of delta (δ, 1–4 Hz), theta (θ, 4–8 Hz), alpha (α, 8–13 Hz), low beta (β1, 13–18 Hz), high beta (β2, 18–25 Hz), low gamma (γ1, 25–50 Hz), and high gamma (γ2, 50–100 Hz). To capture the concurrent changes in both low- and high-frequency ranges, we calculated the ratio of the total power in the alpha, low beta, and high beta bands over the power in high gamma band (i.e., (α + β1 + β2)/γ2). This ratio represents the relative distribution of power between the low- and high-frequency bands and importantly, is independent from the use of PSD or relative PSD. We discovered that the power ratio (α + β1 + β2)/γ2 was higher in control ASO-treated *Ube3a$^{m\Delta e6/p+}$* mice than control ASO-treated WT mice across all timepoints (*Figure 5E, I*). Similar phenotypes were also observed in the somatosensory cortex (*Figure 5—figure supplement 3*), but the EEG rhythmic activity in the visual cortex was not significantly altered in control ASO-treated *Ube3a$^{m\Delta e6/p+}$* mice (*Figure 5—figure supplement 4*). These results indicate that maternal *Ube3a* deficiency alters EEG rhythms in the frontal and somatosensory cortices, and the power ratio (α + β1 + β2)/γ2 can be a robust measure of the effects of Ube3a-as and Snord115 ASOs.

Treating *Ube3a$^{m\Delta e6/p+}$* mice with Ube3a-as ASO at the juvenile age caused a decrease of the power in the alpha, low beta, and high beta bands and an increase of the power in the high gamma band, particularly at 3 and 6 weeks post ASO injection (*Figure 5B–D*; *Figure 5—figure supplement 2A*; *Figure 5—figure supplement 3A, B*), which led to the normalization of the ratio (α + β1 + β2)/γ2 in the frontal and somatosensory cortices, as the ratios in Ube3a-as ASO-treated *Ube3a$^{m\Delta e6/p+}$* mice were indistinguishable from those in control ASO-treated WT mice (*Figure 5E*; *Figure 5—figure supplement 3C*). In contrast, Snord115 ASO only showed such effects in the frontal cortex at 3 weeks post ASO injection, and the effects waned at later timepoints (*Figure 5E*; *Figure 5—figure supplement 3C*). This difference between Ube3a-as ASO and Snord115 ASO generally correlates with their difference in upregulating Ube3a proteins (see below). When *Ube3a$^{m\Delta e6/p+}$* mice were treated with ASOs at the adult age, Ube3a-as ASO and Snord115 ASO were also able to reduce the power in the low frequency bands and increased the power in the high gamma band in the frontal cortex (*Figure 5F–H*; *Figure 5—figure supplement 2B*), thereby reducing the ratio (α + β1 + β2)/γ2 (*Figure 5I*). These effects also waned over time, consistent with the change of Ube3a protein levels (*Figure 3F*). Altogether, these results show that upon reactivation of the paternal *Ube3a* by *Ube3a-ATS*-targeted ASOs in juvenile or adult *Ube3a$^{m\Delta e6/p+}$* mice, the abnormal EEG rhythmic activity in *Ube3a$^{m\Delta e6/p+}$* mice can be reversed in a Ube3a protein level-dependent manner.

## Reactivation of paternal *Ube3a* expression restores normal sleep pattern in *Ube3a$^{m\Delta e6/p+}$* mice

To study the sleep architecture, we used the EEG and EMG signals and a convolutional neural network-based algorithm SPINDLE (*Miladinović et al., 2019*) to classify the brain states into rapid eye movement (REM) sleep, non-rapid eye movement (NREM) sleep, and wake throughout 24 hr

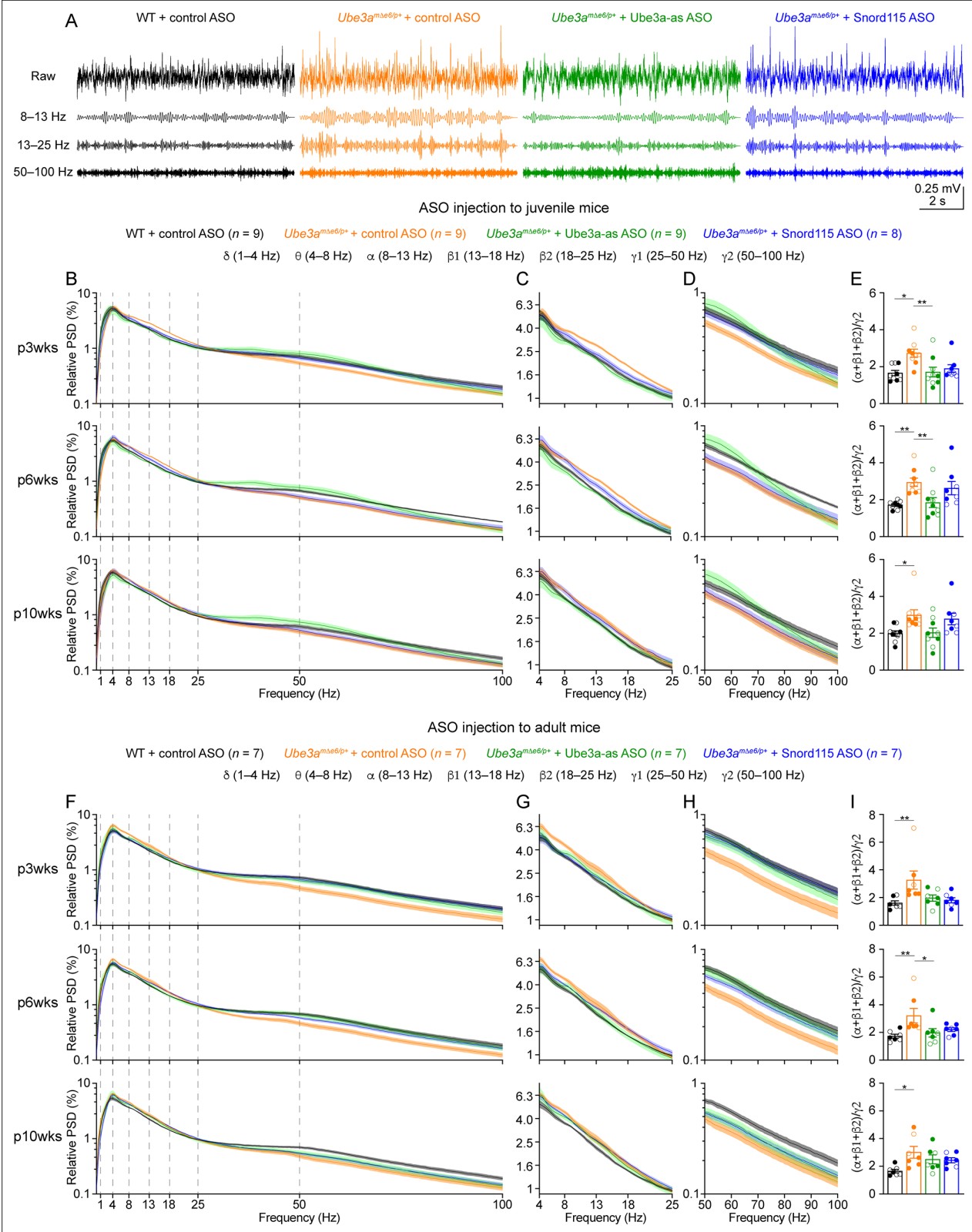

**Figure 5.** Reactivation of paternal *Ube3a* rescues abnormal electroencephalogram (EEG) rhythms in *Ube3a^{mΔe6/p+}* mice. (**A**) Juvenile wild-type (WT) mice were injected with control antisense oligonucleotide (ASO) and *Ube3a^{mΔe6/p+}* mice with control, Ube3a-as, or Snord115 ASO. Representative raw EEG traces and their band-pass filtered traces from the left frontal cortices at 6 weeks post ASO injection. (**B–D**) Relative EEG power spectral density (PSD) curves from the left front cortices at 3, 6, and 10 weeks post ASO injection indicated by p3wks, p6wks, and p10wks, respectively. The dashed

*Figure 5 continued on next page*

*Figure 5 continued*

lines indicate different frequency bands (**B**). The expanded theta ($\theta$)–high beta (β2) and high gamma (γ2) bands are shown in (**C**) and (**D**). Lines and shades are mean and standard error of the mean (SEM), respectively. Control ASO-treated *Ube3a*$^{m\Delta e6/p+}$ mice show an increase of power in the 8–25 Hz range and a decrease of power in the 50–100 Hz range as compared to control ASO-treated WT mice. Ube3a-as ASO reduces the power in 8–25 Hz and increases the power in 50–100 Hz in *Ube3a*$^{m\Delta e6/p+}$ mice. Snord115 ASO has a similar effect in *Ube3a*$^{m\Delta e6/p+}$ mice at 3 weeks post ASO injection. See *Figure 5—figure supplement 2* for statistical comparisons. (**E**) Summary data show the ratio of power in 8–25 over 50–100 Hz. Control ASO-treated *Ube3a*$^{m\Delta e6/p+}$ mice show a higher ratio than control ASO-treated WT mice. Ube3a-as ASO reduces the ratio in *Ube3a*$^{m\Delta e6/p+}$ mice. Snord115 ASO has a similar effect at 3 weeks post ASO injection. (**F–I**) Similar to (**B–E**), but for ASO injection into adult mice. Note, Ube3a-as ASO reduces the ratio of power in 8–25 Hz over 50–100 Hz in *Ube3a*$^{m\Delta e6/p+}$ mice. The numbers of tested mice are indicated in the figure. Each filled (male) or open (female) circle represents one mouse. Bar graphs are mean ± SEM. Kruskal–Wallis test with Dunn's multiple comparison test (**E, I**) for all pairs of groups, *$p < 0.05$, **$p < 0.01$.

The online version of this article includes the following figure supplement(s) for figure 5:

**Figure supplement 1.** Absolute power spectral densities (PSDs) of electroencephalogram (EEG) signals from different brain regions.

**Figure supplement 2.** Reactivation of paternal *Ube3a* in *Ube3a*$^{m\Delta e6/p+}$ mice restores the relative power in different electroencephalogram (EEG) frequency bands.

**Figure supplement 3.** Reactivation of paternal *Ube3a* rescues abnormal electroencephalogram (EEG) rhythms in the somatosensory cortex of *Ube3a*$^{m\Delta e6/p+}$ mice.

**Figure supplement 4.** Normal electroencephalogram (EEG) rhythms in the visual cortex of *Ube3a*$^{m\Delta e6/p+}$ mice.

(*Figure 6—figure supplement 1A*). The difference in the EEG PSDs did not affect the accuracy of SPINDLE (*Figure 6—figure supplement 1B, C*). Overall, mice spent more time in REM and NREM sleep and less time in wake during the light phase than the dark phase (*Figure 6*). The time in wake was similar between control ASO-treated WT and *Ube3a*$^{m\Delta e6/p+}$ mice across ages (*Figure 6C, F*). Control ASO-treated *Ube3a*$^{m\Delta e6/p+}$ mice spent significantly less time in REM sleep than control ASO-treated WT mice in the light phase, but this phenotype was more variable in adult mice (*Figure 6A, D*). Correspondingly, control ASO-treated *Ube3a*$^{m\Delta e6/p+}$ mice spent slightly more time in NREM sleep than control ASO-treated WT mice because REM sleep is a small fraction of the total sleep (*Figure 6B, E*). Thus, the sleep disturbance in *Ube3a*$^{m\Delta e6/p+}$ mice manifests as a selective reduction in REM sleep, which recapitulates the observation in Angelman patients (*Miano et al., 2004*; *Miano et al., 2005*).

Administering Ube3a-as ASO or Snord115 ASO to juvenile *Ube3a*$^{m\Delta e6/p+}$ mice increased their time in REM sleep at 3 and 6 weeks post ASO injection, thereby normalizing their sleep pattern, as their time in REM sleep was indistinguishable from that in control ASO-treated WT mice (*Figure 6A*). This effect was reduced at 10 weeks post ASO injection (*Figure 6A*). When *Ube3a*$^{m\Delta e6/p+}$ mice were treated with ASOs at the adult age, Ube3a-as ASO and Snord115 ASO were less effective in restoring REM sleep (*Figure 6D*). Overall, these results indicate that the sleep disturbance in *Ube3a*$^{m\Delta e6/p+}$ mice can be rescued by reactivation of the paternal *Ube3a* in juvenile mice.

## Partial restoration of Ube3a protein levels does not suppress cortical hyperexcitability in *Ube3a*$^{m\Delta e6/p+}$ mice

Most Angelman syndrome patients develop epileptic seizures within the first 3 years of age (*Williams et al., 2006*; *Bird, 2014*). Although maternal *Ube3a* knockout mice or rats do not develop spontaneous seizures, they exhibit cortical hyperexcitability and epileptiform activity, manifesting as numerous poly-spikes (*Mandel-Brehm et al., 2015*; *Born et al., 2017*; *Born et al., 2021*). Indeed, control ASO-treated *Ube3a*$^{m\Delta e6/p+}$ mice showed significantly more poly-spikes in the frontal and somatosensory cortices than control ASO-treated WT mice (*Figure 7A–C*; *Figure 7—figure supplement 1A, C*). Interestingly, the visual cortices of control ASO-treated *Ube3a*$^{m\Delta e6/p+}$ mice did not exhibit this epileptiform activity (*Figure 7—figure supplement 1B, D*). Ube3a-as ASO or Snord115 ASO treatment of juvenile *Ube3a*$^{m\Delta e6/p+}$ mice did not significantly reduce poly-spikes as compared to control ASO, although Snord115 ASO-treated *Ube3a*$^{m\Delta e6/p+}$ mice showed a 50% reduction in the number of poly-spikes (*Figure 7B*; *Figure 7—figure supplement 1A*). Similarly, treating adult *Ube3a*$^{m\Delta e6/p+}$ mice with Ube3a-as ASO or Snord115 ASO did not cause a significant decrease of poly-spikes at any timepoint (*Figure 7C*; *Figure 7—figure supplement 1C*). Thus, these results indicate that under our experimental conditions where Ube3a protein levels are partially restored in juvenile or adult *Ube3a*$^{m\Delta e6/p+}$ mice, their cortical hyperexcitability phenotype cannot be reversed.

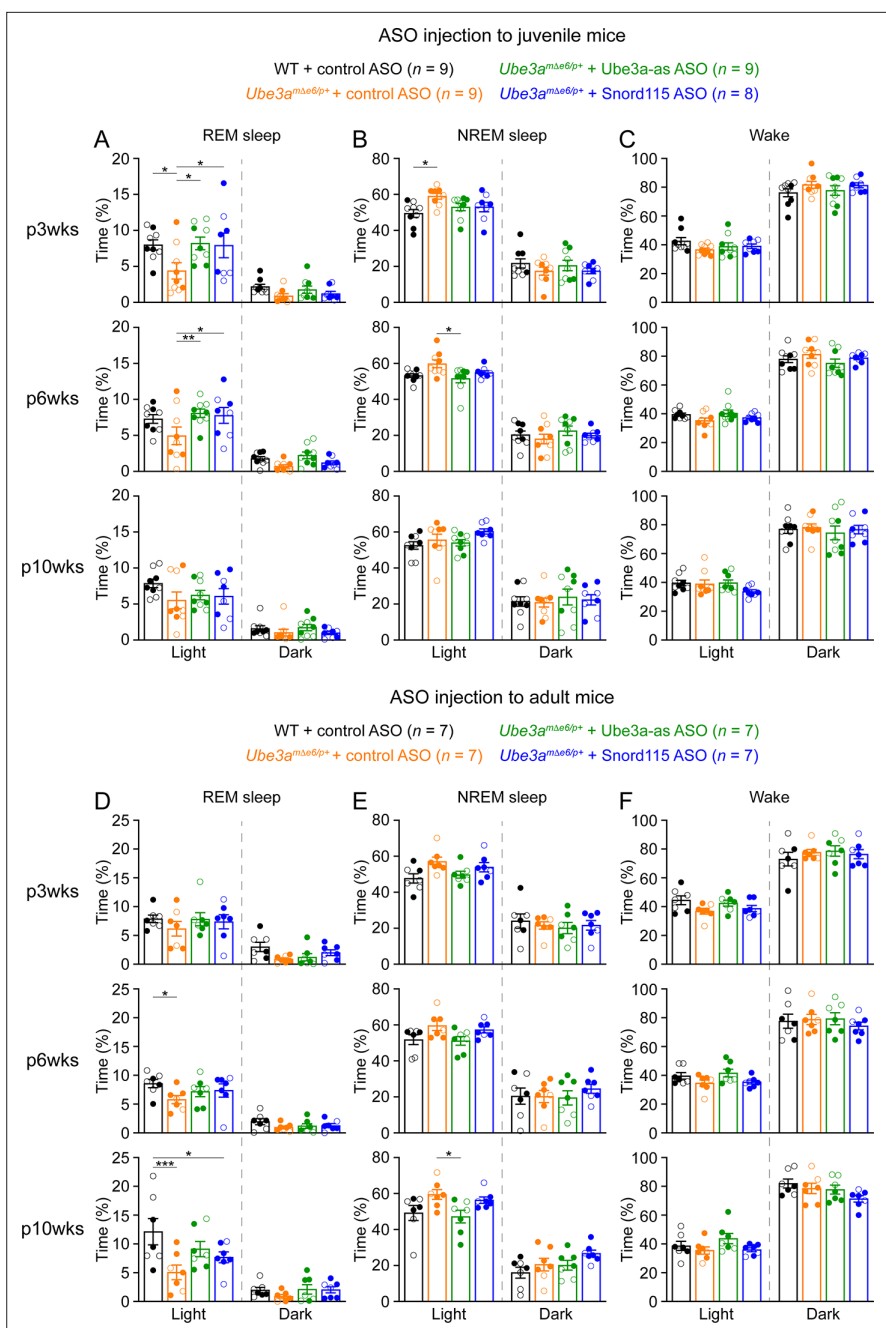

**Figure 6.** Reactivation of paternal *Ube3a* rescues abnormal rapid eye movement (REM) sleep in *Ube3a^{mΔe6/p+}* mice. (**A**) Summary data of the cumulative REM sleep time at 3, 6, and 10 weeks post antisense oligonucleotide (ASO) injection into juvenile mice. Control ASO-treated *Ube3a^{mΔe6/p+}* mice spend less time in REM sleep than control ASO-treated wild-type (WT) mice. Both Ube3a-as and Snord115 ASOs improve REM sleep in *Ube3a^{mΔe6/p+}* mice. (**B, C**) Similar to (**A**), but for non-rapid eye movement (NREM) sleep (**B**) and wake (**C**). (**D–F**) Similar to (**A–C**), but for ASO injection into adult mice. The rescue effect of Ube3a-as and Snord115 ASOs is reduced as compared to ASO injection into juvenile mice. The numbers of tested mice are indicated in the figure. Each filled (male) or open (female) circle represents one mouse. Bar graphs are mean ± standard error of the mean (SEM). Two-way analysis of variance (ANOVA) with Tukey multiple comparison test for all pairs of groups, *p < 0.05, **p < 0.01, ***p < 0.001.

The online version of this article includes the following figure supplement(s) for figure 6:

**Figure supplement 1.** Validation of sleep staging by SPINDLE program.

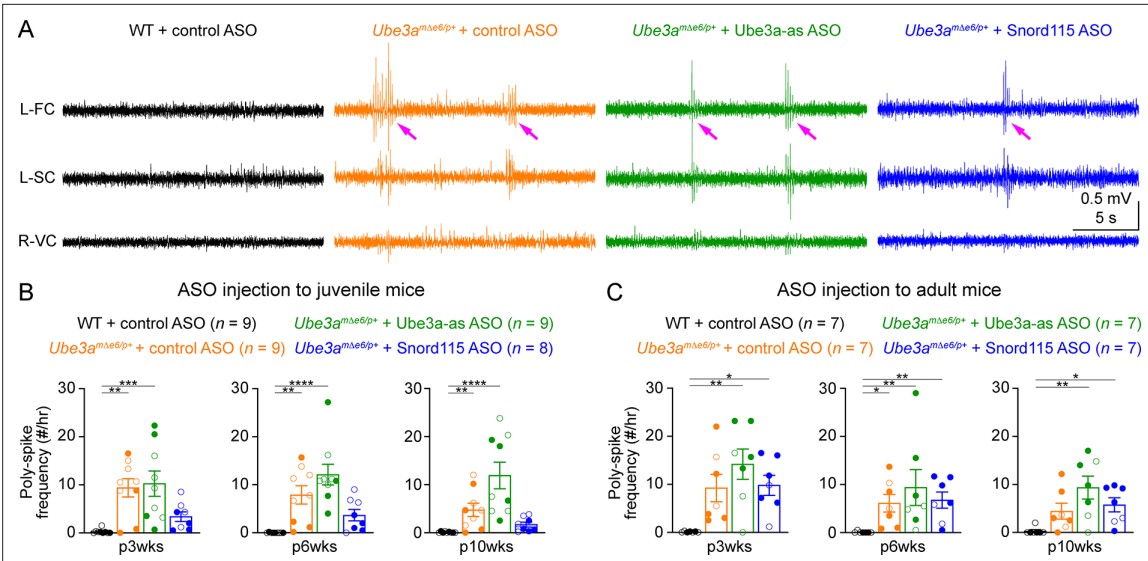

**Figure 7.** Reactivation of paternal *Ube3a* does not suppress poly-spikes in *Ube3a^{mΔe6/p+}* mice. (**A**) Representative electroencephalogram (EEG) traces from the left frontal cortices (L-FC), left somatosensory cortices (L-SC), and right visual cortices (R-VC) at 3 weeks post antisense oligonucleotide (ASO) injection into juvenile mice. (**B**) Summary data showing the frequencies of poly-spikes from the left frontal cortices at 3, 6, and 10 weeks post ASO injection into juvenile mice. Control ASO-treated *Ube3a^{mΔe6/p+}* mice show many more poly-spikes than control ASO-treated wild-type (WT) mice. Snord115 ASO modestly reduces poly-spikes, whereas Ube3a-as ASO does not. (**C**) Similar to (**B**), but for ASO injection into adult mice. Neither Ube3a-as nor Snord115 reduces poly-spikes. The numbers of tested mice are indicated in the figure. Each filled (male) or open (female) circle represents one mouse. Bar graphs are mean ± standard error of the mean (SEM). Kruskal–Wallis test with Dunn's multiple comparison test for all pairs of groups, $*p < 0.05$, $**p < 0.01$, $***p < 0.001$, $****p < 0.0001$.

The online version of this article includes the following figure supplement(s) for figure 7:

**Figure supplement 1.** Poly-spikes in the somatosensory and visual cortices of *Ube3a^{mΔe6/p+}* mice.

## Modulation of EEG rhythms and REM sleep by ASOs tracks the Ube3a protein levels in *Ube3a^{mΔe6/p+}* mice

Our results above show that Ube3a-as ASO and Snord115 ASO upregulate Ube3a proteins (*Figure 3*; *Figure 3—figure supplement 2*) and modulate EEG rhythms (*Figure 5*) and REM sleep (*Figure 6*) to different extents in *Ube3a^{mΔe6/p+}* mice depending on the age of ASO injection and post injection time. To understand how well the modulation of EEG rhythms and REM sleep by the ASOs reflects the Ube3a protein levels, we determined the relationships between Ube3a protein levels and EEG relative power in different frequency bands, power ratio $(\alpha + \beta1 + \beta2)/\gamma2$, or time in REM sleep. We first averaged the Ube3a protein levels across different brain regions to estimate the overall Ube3a protein levels in each mouse from the Western blot experiments (*Figure 3*; *Figure 3—figure supplement 2*) and then obtained the mean Ube3a protein levels at each of the 3-, 6-, and 10-week timepoints post ASO injection. Since the Ube3a protein levels in *Ube3a^{mΔe6/p+}* mice were expressed a fraction of the WT levels, for the corresponding EEG relative power in different frequency bands, power ratio $(\alpha + \beta1 + \beta2)/\gamma2$, and time in REM sleep, we also normalized the data by the means of those in the corresponding control ASO-treated WT mice. For both ASO injection into juvenile and adult mice, the relative power in the theta ($\theta$, 4–8 Hz), alpha ($\alpha$, 8–13 Hz), low beta ($\beta1$, 13–18 Hz), and high beta ($\beta2$, 18–25 Hz) bands negatively correlated with the Ube3a protein levels, whereas the relative power in the low gamma ($\gamma1$, 25–50 Hz) and high gamma ($\gamma2$, 50–100 Hz) bands positively correlated with the Ube3a protein levels. Therefore, the power ratio $(\alpha + \beta1 + \beta2)/\gamma2$ also negatively correlated with the Ube3a protein levels (*Figure 8A*). However, the relative power in the delta ($\delta$, 1–4 Hz) band did not correlate with the Ube3a protein levels (*Figure 8A*). Finally, the time in REM sleep during the light phase positively correlated with the Ube3a protein levels too (*Figure 8B*). In the dark phase, the positive correlation between the time in REM sleep and the Ube3a protein levels existed only for the ASO injection into the juvenile mice (*Figure 8B*). Hence, these results indicate that the relative power

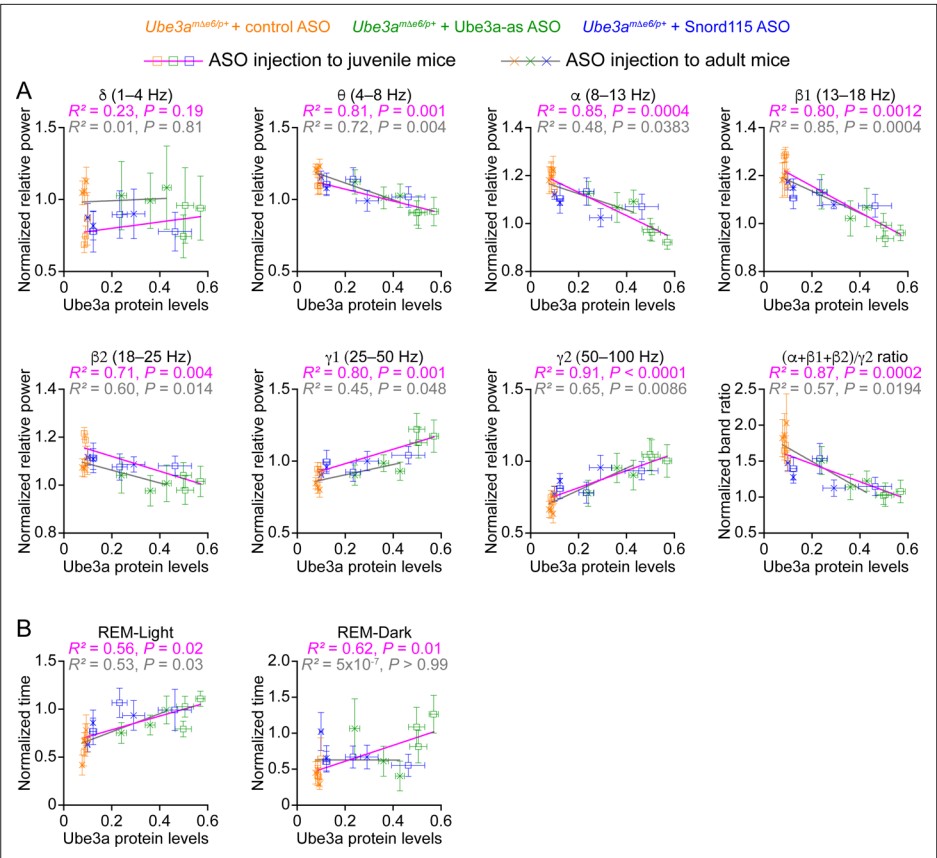

**Figure 8.** Electroencephalogram (EEG) rhythms and rapid eye movement (REM) sleep correlate with Ube3a protein levels in *Ube3a^{mΔe6/p+}* mice. (**A**) The relationships between the Ube3a protein levels and EEG relative power from the frontal cortices of *Ube3a^{mΔe6/p+}* mice injected with control, Ube3a-as, or Snord115 antisense oligonucleotide (ASO) across 3, 6, and 10 weeks post ASO injection into juvenile and adult mice. The Ube3a protein levels from all brain regions were averaged for each mouse. The relative EEG power within each frequency band and the power ratio ($\alpha$ + $\beta$1 + $\beta$2)/$\gamma$2 were normalized by the means of those in wild-type (WT) mice injected with control ASO. The relationships were fitted with a linear regression ($Y = aX + b$; $X$, Ube3a protein levels; $Y$, normalized EEG relative power or power ratio ($\alpha$ + $\beta$1 + $\beta$2)/$\gamma$2; $a$, $b$, constants). (**B**) Similar to (**A**), but for the relationships between the Ube3a protein levels and REM sleep time in light and dark phases. Data are mean ± standard error of the mean (SEM). $R^2$ indicates the goodness of fit. p < 0.05 indicates a significant deviation of slope from zero.

of EEG rhythms and time in REM sleep dynamically follow the Ube3a protein levels in *Ube3a^{mΔe6/p+}* mice that are regulated by the ASOs.

## Discussion

Genetic approaches to restoring *UBE3A* expression holds great promise for treating Angelman syndrome because they tackle the disease root cause. Three active clinical trials of *UBE3A-ATS*-targeted ASOs have generated a great deal of excitement and expectation in the community. Meanwhile, an increasing number of studies in mouse models of Angelman syndrome demonstrate that *Ube3a* must be reinstated in late embryonic and early postnatal development to correct most neurological phenotypes (*Silva-Santos et al., 2015*; *Gu et al., 2019*; *Rotaru et al., 2018*; *Sonzogni et al., 2020*). Among the previously tested phenotypes, only a small subset (i.e., synaptic transmission, plasticity, and spatial memory) can be improved upon increasing Ube3a at the age of 6 weeks or older, and slightly a few more (i.e., rotarod performance and susceptibility to seizure induction) when increasing Ube3a at postnatal day 21 (*Supplementary file 1*). Despite limited prior successes in rescuing juvenile and adult maternal *Ube3a* deficiency mice, we chose these two ages to examine the effects of ASO therapy on

cortical hyperexcitability, altered EEG power spectrum, and sleep disturbance because these ages are more translationally relevant than the neonatal period. Our study reveals that a single ICV injection of *Ube3a-ATS*-targeted ASOs to *Ube3a*$^{mΔe6/p+}$ mice, a new rodent model of Angelman syndrome (***Shi et al., 2022***), upregulates Ube3a proteins including the critical short isoform and restores the EEG power spectrum and sleep pattern for at least 6 weeks, particularly upon treatment at the juvenile age. Therefore, our results significantly expand the range of phenotypes that can be reversed by restoring *Ube3a* expression in juvenile and adult mice. Interestingly, we were not able to reduce the frequency of poly-spikes in *Ube3a*$^{mΔe6/p+}$ mice at either age (***Figure 7***). It is possible that suppression of poly-spikes requires upregulation of Ube3a starting at a younger age or reaching to a higher level than what we have achieved, both of which should be tested in future studies. Nevertheless, this result indicates that poly-spikes are independent from the EEG power spectrum and sleep pattern deficits, and probably involve a different mechanism.

A critical finding of our study is that the improvement in the EEG power spectrum and sleep pattern tracks the increase in Ube3a protein levels across different ASOs, injection ages, and timepoints post injection (***Figure 8***). This suggests that following a bolus injection of ASOs, both phenotypes are acutely modulated by the Ube3a levels that decrease over time due to ASO clearance. Future studies should determine if repeated administration of ASOs can generate a long-lasting improvement of the phenotypes beyond the period when Ube3a levels are sufficiently upregulated, as the outcome can help inform the ASO treatment schedule in clinical trials. The ASO treatment in adult mice is less effective than juvenile mice, which is consistent with a recent finding that treating *Ube3a*$^{mΔe5/p+}$ mice with Ube3a-as ASO around P35 modestly reduced the EEG PSD in the delta frequency band (***Spencer et al., 2022***). This could be due to two reasons. First, the reversibility of disturbed EEG power spectrum and sleep may decrease over age, just like other neurological deficits (***Supplementary file 1***). Second, the ASO treatment in adult mice causes a smaller increase of Ube3a protein than juvenile mice (***Figure 3***). Given the strong correlation between Ube3a levels and phenotypic improvement, we speculate that the latter is more likely the reason, and a higher dose of ASO or an ASO with a higher efficacy in downregulating *Ube3a-ATS* should further increase Ube3a protein and improve these two phenotypes in adult mice.

Previous EEG studies of *Ube3a*$^{mΔe5/p+}$ mice emphasized an increase of absolute power or PSD in the delta frequency band as compared to WT mice, but the results varied among studies (***Ehlen et al., 2015***; ***Born et al., 2017***; ***Sidorov et al., 2017***; ***Copping and Silverman, 2021***). One study recorded local field potential in layer 4 of the primary visual cortex from awake mice that were head-fixed and viewing a static gray screen. The absolute delta (2–4 Hz) power of local field potential was increased when *Ube3a*$^{mΔe5/p+}$ mice were on a 129 strain background, but not on a C57BL/6 background. Interestingly, a reduction of the relative power in the gamma band (30–50 Hz) was observed in both strains (***Sidorov et al., 2017***). In contrast, two other studies reported an increase in delta (0–4 or 0.5–4 Hz) power of chronic EEG recorded from the cortical surface of freely moving *Ube3a*$^{mΔe5/p+}$ mice on a C57BL/6 background (***Born et al., 2017***; ***Copping and Silverman, 2021***), but not on a 129 strain background (***Born et al., 2017***). Finally, it was reported that the relative delta power of cortical surface EEG was reduced in *Ube3a*$^{mΔe5/p+}$ mice on the C57BL/6J background during NREM sleep in the night, but not in the day (***Ehlen et al., 2015***). Our absolute PSD results from the new *Ube3a*$^{mΔe6/p+}$ mice are qualitatively similar to the previous results (***Figure 5— figure supplement 1***), but relative PSD analysis reveals an increase of relative power in the theta, alpha, or beta frequency bands and a decrease in the low or high gamma frequency bands. These differences could be due to different mutations, genetic backgrounds, mouse ages, experimental conditions, or different brain states included in the analyses. Nevertheless, we observed a robust and consistent increase in the power ratio (α + β1 + β2)/γ2 across timepoints (***Figure 5***). In fact, inspection of previous results suggests a common pattern that EEG power is relatively higher in maternal *Ube3a* deficiency mice than WT mice in the lower frequency bands (i.e., beta or lower) and relatively lower in the gamma bands, although the results were not always statistically significant (***Born et al., 2017***; ***Sidorov et al., 2017***; ***Copping and Silverman, 2021***). Similarly, the EEG power spectrum of Angelman syndrome patients also shows such a pattern (***Sidorov et al., 2017***). Thus, we propose that a power ratio between the low- and high-frequency bands would be a more robust measure of the EEG power spectrum phenotype in Angelman syndrome and its mouse models.

The EEG power spectrum and sleep phenotypes of maternal *Ube3a* knockout mice directly correlate with those in Angelman syndrome patients, but about 70% of Angelman syndrome patients have a larger deletion in the chromosomal region 15q11–q13 that encompasses *UBE3A* and other genes with biallelic or paternal expression (*Bird, 2014*; *Buiting et al., 2016*). These deletion Angelman syndrome patients have more severe clinical phenotypes (*Williams et al., 2006*; *Bird, 2014*; *Buiting et al., 2016*) and a larger increase in the EEG theta band power (*Frohlich et al., 2019*) than non-deletion patients, suggesting that the heterozygosity of those biallelically expressed genes contribute to the severity of Angelman syndrome. Thus, future studies are necessary to determine the contribution of these genes to the neurological phenotypes including disturbed EEG power spectrum and sleep. Particularly, the *GABRB3*, *GABRA5*, and *GABRG3* genes in this region encode $GABA_A$ receptor subunit β3, α5, and γ3, respectively, and are likely to be critical given the vital role of GABAergic inhibition in brain functions and the fact that heterozygous loss-of-function *GABRB3* variants cause a broad spectrum of epilepsies (*Møller et al., 2017*). The *HERC2* gene encoding HECT domain and RCC1-like domain 2 can also be important because this E3 ubiquitin protein ligase interacts with UBE3A (*Kühnle et al., 2011*) and its loss-of-function causes a neurodevelopmental disorder with Angelman syndrome-like features (*Puffenberger et al., 2012*; *Harlalka et al., 2013*). Since *UBE3A-ATS*-targeted ASOs do not affect these biallelically expressed genes, it will be of great importance to determine to what extent reactivating paternal *Ube3a* can rescue the EEG power spectrum and sleep phenotypes in a large deletion mouse model (*Jiang et al., 2010*).

Our study has several translational implications for the ASO and other clinical trials of Angelman syndrome. First, even though it is unclear how the mouse developmental stages in which the neurological phenotypes can be reversed are related to the treatment window for Angelman syndrome, our results suggest that Angelman syndrome patients at different ages may all benefit from the ASO treatment of these two core disease symptoms. Second, the robust correlation between EEG power spectrum and Ube3a levels supports the notion that EEG power spectrum can serve as a quantitative biomarker in clinical trials (*Sidorov et al., 2017*; *Frohlich et al., 2019*; *Hipp et al., 2021*; *Ostrowski et al., 2021*). Third, since the EEG power of Angelman syndrome patients correlates with their symptom severity, particularly the cognitive function (*Hipp et al., 2021*; *Ostrowski et al., 2021*), it is reasonable to speculate that ASO treatment may also improve the cognitive function of Angelman syndrome patients. Finally, clinicians and caregivers consider sleep disturbance as one of the most challenging symptoms and important focuses for new treatment (*Willgoss et al., 2021*). Thus, if ASO therapy can reduce sleep disturbance, then it will improve quality of life for both Angelman syndrome patients and caregivers.

# Materials and methods

## Key resources table

| Reagent type (species) or resource | Designation | Source or reference | Identifiers | Additional information |
|---|---|---|---|---|
| Genetic reagent (*Mus musculus*) | *Ube3a*$^{\Delta e6}$ | This paper | | The new *Ube3a* KO allele that deletes exon 6 |
| Genetic reagent (*Mus musculus*) | B6.129S7-*Ube3a*$^{tm1Alb}$/J | The Jackson Laboratory | RRID:IMSR_JAX:016590 | The previous *Ube3a* KO allele that deletes exon 5, *Ube3a*$^{\Delta e5}$ in the paper |
| Genetic reagent (*Mus musculus*) | B6.129S7-*Ube3a*$^{tm2Alb}$/J | The Jackson Laboratory | RRID:IMSR_JAX:017765 | *Ube3a*$^{YFP}$ in the paper |
| Genetic reagent (*Mus musculus*) | C57BL/6J | The Jackson Laboratory | RRID:IMSR_JAX:000664 | |
| Sequence-based reagent | Control ASO | Ionis Pharmaceuticals, *Meng et al., 2015* | | 5′-CCToAoToAoGGACTATCCAoGoGAA-3′ |
| Sequence-based reagent | Ube3a-as ASO | Ionis Pharmaceuticals, *Meng et al., 2015* | | 5′-CCoAoGoCoCTTGTTGGATAoToCAT-3′ |
| Sequence-based reagent | Snord115 ASO | Ionis Pharmaceuticals, this paper | | 5′-TToGoToAoAGCATCAAAGToAoTGA-3′ |

*Continued on next page*

*Continued*

| Reagent type (species) or resource | Designation | Source or reference | Identifiers | Additional information |
|---|---|---|---|---|
| Antibody | Rabbit monoclonal anti-GFP | Invitrogen | Cat. #: G10362 | IF (1:2000) |
| Antibody | Mouse monoclonal anti-E6AP | Sigma-Aldrich | Cat. #: E8655 | WB (1:1000) |
| Antibody | Mouse monoclonal anti-Gapdh | Proteintech | Cat. #: 60004-1-Ig | WB (1:10,000) |
| Antibody | Mouse monoclonal anti-β3 tubulin | Proteintech | Cat. #: 66240-1-Ig | WB (1:50,000) |
| Antibody | Goat polyclonal anti-rabbit secondary antibody conjugated with Alexa Flour 647 | Invitrogen | Cat. #: A21245 | IF (1:1000) |
| Antibody | Goat polyclonal anti-mouse antibody conjugated with IRDye 800CW | LI-COR Bioscience | Cat. #: 925-32210 | WB (1:20,000) |
| Other | NeuroTrace 435/455 blue fluorescent Nissl stain | Invitrogen | Cat. #: N21479 | IF (1:200) |
| Other | Bolt Bis-Tris Plus 10% mini protein gels | Thermo Fisher Scientific | Cat. #: NW00107BOX | SDS–PAGE for total Ube3a |
| Other | Novex Tris-Glycine 4–12% mini protein gels | Thermo Fisher Scientific | Cat. #: XP04120BOX | SDS–PAGE for separation of Ube3a isoforms |
| Software, algorithm | LAS X software | Leica | RRID:SCR_013673 | Version 3.3.0.16799 |
| Software, algorithm | ImageJ | NIH | RRID:SCR_003070 | Version 1.53c |
| Software, algorithm | LABKIT | *Arzt et al., 2022* | | Version 0.3.9, https://imagej.net/plugins/labkit/ |
| Software, algorithm | Image Studio Lite | LI-COR Biosciences | RRID:SCR_013715 | Version 5.2 |
| Software, algorithm | Sirenia | Pinnacle Technology | RRID:SCR_016184 | Version 1.8.2 |
| Software, algorithm | SPINDLE | *Miladinović et al., 2019* | | https://sleeplearning.ethz.ch |
| Software, algorithm | Prism | GraphPad Software | RRID:SCR_002798 | Version 9 |
| Software, algorithm | Python | Python Software Foundation | RRID:SCR_008394 | Version 3.9.12 |

## Mice

The new *Ube3a* null allele was generated by CRISPR/Cas9-mediated deletion of exon 6. WT *Cas9* mRNA (100 ng/µl) and two sgRNAs (10 ng/µl each) targeting the genomic sequences of *Ube3a* intron 5 (5'-TTACATACCAGTACATGTCTTGG-3') and intron 6 (5'-TGCTTTCTACCAACTGAGACAGG-3') were microinjected into WT C57BL/6J (JAX # 000664) zygotes. Founder mice carrying the exon 6 deletion (*Ube3a*$^{\Delta e6}$) were identified by PCR using a pair of primers (5'-TTGAGAACAATGCAAAGGAAAATGA-3' and 5'-GAGCAAACTGCTGTAGACCC-3') for the WT allele (747 bp) and a pair of primers (5'-TTGA GAACAATGCAAAGGAAAATGA-3' and 5'-TGAGGCTGGCTTTCAAGATTCA-3') for the *Δe6* (314 bp) allele. Founder mice were then backcrossed to WT C57BL/6J mice to generate N1 mice. N1 mice carrying the *Δe6* allele were identified using the same PCR above. Sequencing of the identified N1 mice confirmed that the sequence chr7:59,275,513–59,277,423 were deleted. *Ube3a*$^{\Delta e6}$ mice were

backcrossed to WT C57BL/6J mice for at least five generations prior to experiments. Heterozygous female mice carrying the mutation on their paternal chromosome (*Ube3a^{m+/pΔe6}*) were crossed with WT C57BL/6J mice to generate WT and maternal knockout of *Ube3a* mice (*Ube3a^{mΔe6/p+}*).

*Ube3a^{Δe5}* mice (*Ube3a^{tm1Alb}*, JAX # 016590) were described previously (*Jiang et al., 1998*), in which exon 5 was deleted. Heterozygous female *Ube3a^{m+/pΔe5}* mice were crossed with WT C57BL/6J mice to generate WT and *Ube3a^{mΔe5/p+}* mice. Heterozygous male and female *Ube3a^{m+/pΔe5}* mice were crossed with each other to generate WT and *Ube3a^{mΔe5/ mΔe5}* mice. *Ube3a^{YFP}* mice (*Ube3a^{tm2Alb}*, JAX #017765) were described previously (*Dindot et al., 2008*) and carry a *Ube3a* knockin allele with a yellow fluorescent protein (YFP) fused to the C terminus of Ube3a. Heterozygous male or female *Ube3a^{m+/pYFP}* mice were crossed with WT C57BL/6J mice to obtain heterozygous paternal or maternal *Ube3a^{YFP}* mice (*Ube3a^{m+/pYFP}* or *Ube3a^{mYFP/p+}*), respectively. *Ube3a^{Δe5}* and *Ube3a^{YFP}* mice were also maintained on the C57BL/6J background.

Mice were housed in an Association for Assessment and Accreditation of Laboratory Animal Care International-certified animal facility on a 14/10 hr light/dark cycle. All procedures to maintain and use mice were performed in strict accordance with the recommendations in the Guide for the Care and Use of Laboratory Animals of the National Institutes of Health and were approved by the Institutional Animal Care and Use Committee at Baylor College of Medicine (protocol AN-6544).

## Reverse transcription droplet digital PCR

Mice were anesthetized and decapitated. Brain and liver tissues were extracted and homogenized with Trizol (Thermo Fisher, catalog #15596026), followed by RNase-free DNase treatment (Qiagen, catalog #79254). RNAs were purified with RNeasy Plus Mini Kit (Qiagen, catalog #74136) and reverse transcribed to cDNA by High-Capacity cDNA Reverse Transcription Kit (Thermo Fisher, catalog #4368814). The cDNA concentration was determined by a Nanodrop (Thermo Fisher). Droplet digital PCR (ddPCR) was prepared by mixing the following reagents in a total volume of 20 μl: 2× QX200 ddPCR EvaGreen supermix (Bio-Rad, catalog #1864036, 10 μl), forward and reverse primers (10 μM, 0.4 μl each), cDNA template (10–100 ng, 1 μl), and nuclease-free H$_2$O (8.2 μl). The droplets for ddPCR were generated by a Biorad Automated Droplet Generator (Bio-Rad, catalog #1864101), followed by PCR reaction using a Thermal Cycler C1000 (Bio-Rad). The plate containing the droplets was read by a QX200 Droplet reader (Bio-Rad, catalog #1864001). The primers for detecting different fragments of the *Ube3a* transcripts are provided in *Supplementary file 2*. The expression levels of *Ube3a* transcripts were normalized by the *Gapdh* levels.

## Antisense oligonucleotides

Synthesis and purification of all chemically modified oligonucleotides were performed as previously described (*Swayze et al., 2007*). The 2'-*O*-methoxyethylribose (MOE) gapmer ASOs are 20 nucleotides in length, wherein the central gap segment comprising ten 2'-deoxynucleotides is flanked on the 5' and 3' wings by five 2'-MOE modified nucleotides. All internucleoside linkages are phosphorothioate linkages, except the ones shown as 'o' in the sequences which are phosphodiester. The sequences of the ASOs are as follows: control ASO, 5'-CCToAoToAoGGACTATCCAoGoGoGAA-3'; Ube3a-as ASO, 5'-CCoAoGoCoCTTGTTGGATAoToCAT-3'; and Snord115 ASO, 5'-TToGoToAoAGCATCAAAGToAoTGA-3'. Lyophilized ASOs were formulated in phosphate-buffered saline (PBS) without Ca$^{2+}$ and Mg$^{2+}$ (Gibco, catalog # 14190). ASOs were dissolved in PBS to obtain 50 mg/ml concentrations.

## ICV injection of ASOs

Mice were anesthetized with isoflurane (1.5–2.5%) in oxygen (1 l/min). The body temperature was monitored and maintained at 37°C using a temperature controller (ATC-2000, World Precision Instruments). An incision was made along the midline to expose the skull after the head was fixed in a stereotaxic apparatus. Approximately 0.25-mm-diameter craniotomies were performed with a round bur (0.25-mm-diameter) and a high-speed rotary micromotor (EXL-M40, Osada) at the injection site (see below). A bevelled 50-μm-diameter glass pipette was used to inject ASOs into the right lateral ventricle according to one of the following sets of coordinates that were normalized by the distance between Bregma and Lambda (DBL). (1) Anterior/posterior (AP): 0.055 of DBL, medial/lateral (ML): 0.238 of DBL, dorsal/ventral (DV): −0.499 of DBL; (2) AP: 0.055 of DBL, ML: 0.238 of DBL, DV: −0.594

of DBL; (3) AP: 0.071 of DBL, ML: 0.238 of DBL, DV, −0.713 of DBL. The results were similar among these three sets of coordinates and were grouped together. ASO solution was injected at a rate of 407 nl/s using an UltraMicroPump III and a Micro4 controller (VAR-3735, World Precision Instruments). A total of 10 µl ASO solution (50 mg/ml) was administered for a total dosage of 500 µg/mouse except six mice that were injected with 5 µl ASO solution for a total dosage of 250 µg/mouse at the age of 3 weeks and used in the Western blot experiments. The results from these six mice were similar to other mice (*Figure 3—figure supplement 5*) and grouped together. After injection, the pipette was held in place for 10 min before withdrawal. The skin was sutured, and mice were allowed to recover from anesthesia in a cage placed on a heating pad. When the recovery takes longer than 1 hr, the duration on the heating pad should not exceed 1 hr, as longer exposure of mice on the heating pad significantly reduces post-surgery survival rates (less than 1 hr: 1 out of 102 injected mice died, more than 1 hr: 31 out of 106 injected mice died, p < 0.0001).

## Immunohistochemistry and fluorescent microscopy

Mice were anesthetized and transcardially perfused with PBS (pH 7.4) followed by 4% paraformalde-hyde in PBS (pH 7.4). Brains were then post-fixed for overnight in 4% paraformaldehyde at 4°C and sectioned into 40-µm sagittal slices using a vibratome (VT1000S, Leica). Brain sections were incubated in blocking solution (0.2% Triton X-100 in PBS with 5% normal goat serum) for 1 hr at 4°C and then with a primary rabbit monoclonal anti-GFP antibody (Invitrogen, catalog # G10362, lot # 1965886, 1:2000 dilution) that recognizes YFP for overnight at 4°C. Sections were washed with 0.2% Triton X-100 in PBS and then incubated with a goat polyclonal anti-rabbit secondary antibody conjugated with Alexa Flour 647 (Invitrogen, catalog # A21245, lot # 1623067, 1:1000 dilution) in blocking solution for 3 hr at room temperature. After antibody staining, sections were incubated with NeuroTrace 435/455 blue fluorescent Nissl stain (Invitrogen, catalog # N21479, 1:200 dilution) in 0.2% Triton X-100 in PBS at room temperature for 1 hr to label neurons. Sections were washed with 0.2% Triton X-100 in PBS and mounted in ProLong Diamond Antifade Mountant (Invitrogen, catalog # P36961). High-resolution (1024 × 1024) single-plane images of the brain sections were acquired on a TCS SP8X Confocal Microscope (Leica) using a ×20 oil objective (HC PL APO CS2 ×20, NA = 0.75). Mosaic images were stitched together using LAS X software v3.3.0.16799 (Leica) and visualized and exported by ImageJ 1.53c (NIH). To quantify Ube3a-YFP levels, cell somas were identified and segmented based on fluo-rescent Nissl stain using an ImageJ plugin, LABKIT (*Arzt et al., 2022*). For each image, YFP fluores-cence intensity was measured and averaged across all identified cell somas and then normalized by the mean of *Ube3a$^{mYFP/p+}$* mice.

## Western blot and RT-qPCR

Mice were anesthetized and decapitated. The brains were extracted, and different regions were dissected from both hemispheres. The brain tissues from the right hemisphere were used for Western blots and the left hemisphere for RT-qPCR. Tissues were frozen at −80°C until analysis.

For Western blots, the brain tissues were homogenized in RIPA buffer containing 50 mM Tris–HCl (pH 8.0), 150 mM NaCl, 1% Triton X-100, 0.5% Na-deoxycholate, 0.1% sodium dodecyl sulfate (SDS), 1 mM ethylenediaminetetraacetic acid (EDTA), 5% glycerol, and 1 cOmplete Protease Inhibitor tablet (Roche, # SKU 11836170001). After homogenization, tissue debris was removed by centrifugation and protein concentrations were determined by Pierce BCA Protein Assay Kit (Thermo Fisher Scientific, catalog # 23225). To measure total Ube3a, 10 µg of proteins per sample were resolved by SDS–polyacrylamide gel electrophoresis (PAGE) with 10% Bis-Tris gels (Thermo Fisher Scientific, catalog # NW00107BOX) and transferred onto nitrocellulose membranes. To separate the two Ube3a isoforms, 3 µg of proteins per sample were resolved by SDS–PAGE with 4–12% Tris-Glycine gradient gels (Thermo Fisher Scientific, catalog # XP04120BOX) at 200 volts for 90 min. Ube3a was detected by a mouse monoclonal anti-E6AP antibody (Sigma-Aldrich, catalog # E8655, lot # 118M4792V, 1:1000 dilution) that recognizes both Ube3a isoforms 2 and 3. Gapdh was detected by a mouse monoclonal anti-Gapdh antibody (Proteintech, catalog # 60004-1-Ig, lot # 10004129, 1:10,000 dilution). β3 tubulin was detected by a mouse monoclonal anti-β3 tubulin antibody (Proteintech, catalog # 66240-1-Ig, lot # 10004491, 1:50,000 dilution). Primary antibodies were detected by a goat polyclonal anti-mouse antibody conjugated with IRDye 800CW (LI-COR Bioscience, catalog # 925-32210, lot # C90130-03, 1:20,000 dilution). Proteins were visualized and

quantified using an Odyssey CLx Imager and Image Studio Lite version 5.2 (LI-COR Biosciences). To quantify the total Ube3a levels, both Ube3a isoforms 2 and 3 were included. To quantify the levels of each isoform, total Ube3a and isoform 3 were quantified, and the isoform 2 levels were measured by subtracting the isoform 3 levels from the total Ube3a levels. Total Ube3a, Ube3a isoform 2, or Ube3a isoform 3 levels were first normalized by the Gapdh or β3 tubulin levels and then by the average total Ube3a, Ube3a isoform 2, or Ube3a isoform 3 of all WT mice from the same blot, respectively.

For RT-qPCR, the brain tissues were homogenized in RLT buffer (Qiagen, catalog # 79216) containing 1% (vol/vol) β-mercaptoethanol. Homogenization was performed for 20 s at 6000 rpm using a Fast-Prep Automated Homogenizer (MP Biomedicals). Total RNA was then purified using the RNeasy 96 Kit (Qiagen, catalog # 74182) that included an in-column DNA digestion with 50 U of DNase I (Invitrogen, catalog # 18047019). RT-qPCR was performed in triplicate with the EXPRESSS One-Step SuperScript qRT-PCR kit (Thermo Fisher Scientific, catalog # 11781200). Gene-specific primers and probes are provided in *Supplementary file 2*. The expression levels of *Ube3a* or *Ube3a-ATS* were normalized by the *Gapdh* levels and then by the average *Ube3a* or *Ube3a-ATS* levels of all WT mice from the same experiment, respectively.

## Video-EEG and EMG recordings

Video-EEG/EMG recordings were performed as previously described (*Chen et al., 2020*). Briefly, 1 week after ASO injection, mice were anesthetized with 2.5% isoflurane in oxygen, and craniotomies were performed as described above for ICV injection. Perfluoroalkoxy polymer (PFA)-coated silver wire electrodes (A-M Systems, catalog # 786000, 127 mm bare diameter, 177.8 mm coated diameter) were used for grounding at the right frontal cortex, referencing at the cerebellum, and recording at the left frontal cortex (AP: 0.475 of DBL, ML: −0.071 of DBL, DV: −1.5 mm), left somatosensory cortex (AP: −0.190 of DBL, ML: −0.428 of DBL, DV: −1.5 mm), and right visual cortex (AP: −0.808 of DBL, ML: 0.594 of DBL, DV: −1.5 mm). An EMG recording and an EMG reference electrode were inserted into the neck muscles. All electrodes were soldered to an adaptor prior to the surgery. The electrodes and adaptor were secured on the skull by dental acrylic. The skin was sutured and attached to the dried dental acrylic. Mice were singly housed to recover for at least 1 week after the surgeries. Before recording, mice were individually habituated in the recording chambers (10-inch diameter of Plexiglas cylinder) for 24 hr. EEG/EMG signals (5000 Hz sampling rate with a 0.5 Hz high-pass filter) and videos (30 frames/s) were recorded synchronously for more than 48 continuous hours using a 4-channel EEG/EMG tethered system and Sirenia 1.8.2 software (Pinnacle Technology).

## EEG poly-spikes and power spectrum analyses

Poly-spikes and PSD were analyzed from the same 6 hr of each recording (12 AM–1 AM, 4 AM–5 AM, 8 AM–9 AM, 12 PM–1 PM, 4 PM–5 PM, and 8 PM–9 PM on the second day). EEG/EMG traces were visualized in Sirenia Seizure 1.8.2 software (Pinnacle Technology) to identify episodes of poly-spikes and artifacts. An episode of poly-spikes is defined as a cluster of three or more spikes on any of the EEG channels. PSD analyses of EEG data were performed using custom scripts in Python. Prior to PSD calculation, data were detrended by subtracting the mean of the data. The data segments containing artifacts on any of the EEG channels were first excluded, and then an eighth-order Butterworth filter was applied to each channel to bidirectionally notch filter around 60 Hz (±2 Hz bandwidth) to remove power-line noise. The PSDs were then estimated for each channel using a Welch's periodogram (*Welch, 1967*) with a 2 s Hanning window (achieving a frequency resolution of 0.5 Hz) and 50% overlap between windows. To account for the effect of notch filtering, the PSD was linearly interpolated between 58 and 62 Hz using the 10 points before and after the mentioned ranges. To analyze different frequency bands, the PSD was segmented into seven bands: delta (1–4 Hz), theta (4–8 Hz), alpha (8–13 Hz), low beta (13–18 Hz), high beta (18–25 Hz), low gamma (25–50 Hz), and high gamma (50–100 Hz). The power within a frequency band (the area under the PSD curve for a band) was then computed for each band. The relative power in a frequency band is the ratio of the power within the band over the total power within 1–100 Hz. The normalized PSD curves were obtained by dividing the PSD curves with the total power within 1–100 Hz. The low- to high-frequency band ratio

was calculated as the ratio of the total power in the alpha and beta bands (8–25 Hz) over the power in the high gamma band (50–100 Hz).

## Sleep scoring

A convolutional neural network-based algorithm SPINDLE (https://sleeplearning.ethz.ch) was used for automated sleep scoring (*Miladinović et al., 2019*). This method produces domain invariant predictions and makes use of a hidden Markov model to limit state dynamics based on known sleep physiology. Sleep was scored from the entire second day (24 hr) of each recording. EEG signals from the frontal and somatosensory cortices and the EMG signals were used to score each 4 s epoch as wake, NREM sleep, or REM sleep. To assess the performance of this method on our dataset, two WT mice treated with control ASO, two $Ube3a^{m\Delta e6/p+}$ mice with control ASO, and one $Ube3a^{m\Delta e6/p+}$ mouse with Ube3a-as ASO were randomly selected and 1 hr of data from each mouse were scored by SPINDLE and manually by three experts. The Precision, Recall, $F1$-score, and Accuracy were calculated for each pairwise comparison as the following: $Precision = \frac{True\ positive}{True\ positive+False\ positive}$; $Recall = \frac{True\ positive}{True\ positive+False\ negative}$; $F1score = 2 \times \frac{Precision \times Recall}{Precision+Recall}$; $Accuracy = \frac{True\ positive+True\ negative}{True\ positive+True\ negative+False\ positive+False\ negative}$. The accuracy of SPINDLE compared to experts was similar to that of the experts compared among each other (*Figure 6—figure supplement 1*).

## Experimental study design and statistics

Estimation of the sample size was made based on previous studies that used similar assays and pilot experiments. They are within the range that is generally accepted in the field. All experiments were performed and analyzed blind to the genotypes and ASOs. $Ube3a^{m\Delta e6/p+}$ mice were randomly assigned to three groups, each of which received control, Ube3a-as, or Snord115 ASO. Approximately equal number of male and female mice was included in experiments. No data point was excluded.

All reported sample numbers (*n*) represent independent biological replicates that are the numbers of tested mice. Statistical analyses were performed with Prism 9 (GraphPad Software). Student's *t*-test or analysis of variance (ANOVA) with multiple comparison test for all pairs of groups was used to determine if there is a statistically significant difference between two groups or among three or more groups, respectively. One- or two-way ANOVA was applied for one or two independent variables, respectively. Anderson–Darling test, D'Agostino–Pearson, Shapiro–Wilk, and Kolmogorov–Smirnov tests were used to determine if data were normally distributed. Non-parametric Kruskal–Wallis one-way ANOVA with Dunn's multiple comparison test was used for low- to high-frequency band ratio and poly-spike data. The details of all statistical tests, numbers of replicates, and p values are reported in *Supplementary file 3*.

## Materials, data, and code availability

Following the material transfer agreement of Baylor College of Medicine, $Ube3a^{\Delta e6}$ mice will be available upon request or from the Jackson Laboratory (JAX). All data generated or analyzed during this study are included in the manuscript. Computer codes for EEG power spectrum analyses and sleep scoring are available at GitHub (https://github.com/heetkaku/angelman, copy archived at swh:1:rev:a8e5152b011de54a1d2b3ea468a07fe7e2345c5f, *Kaku, 2022*).

## Acknowledgements

We thank Jacob, Debra, and Steven Sukin for inspiring this work, Baylor College of Medicine Genetically Engineered Rodent Models Core led by Dr. Jason Heaney for microinjection of *Cas9* mRNA and sgRNAs to generate $Ube3a^{\Delta e6}$ mice, and Drs. Catherine Chu, Robert Komorowski, James Gilbert, Rodney Samaco, and Huda Zoghbi for discussions. This work was supported in part by the Main Street America Fund at Texas Children's Hospital, the Eunice Kennedy Shriver National Institute of Child Health and Human Development (P50HD103555 to Baylor College of Medicine Intellectual and Developmental Disabilities Research Center, Neurovisualization Core), and the National Cancer Institute (P30CA125123 to Baylor College of Medicine Cancer Center). MX was supported by the National Institute of Neurological Disorders and Stroke (R01NS100893) and the National Institute of Mental Health (R01MH117089) and is a Caroline DeLuca Scholar.

# Additional information

### Competing interests
Armand Soriano, Frank Rigo, Paymaan Jafar-nejad: is a paid employee of Ionis Pharmaceuticals. Arthur L Beaudet: is a paid employee of Luna Genetics. Mingshan Xue: is a consultant to Capsida Biotherapeutics and receives funds from Capsida Biotherapeutics for research not related to this study. The other authors declare that no competing interests exist.

### Funding

| Funder | Grant reference number | Author |
|---|---|---|
| Texas Children's Hospital | Main Street America Fund | Mingshan Xue |
| National Institute of Neurological Disorders and Stroke | R01NS100893 | Mingshan Xue |
| National Institute of Mental Health | R01MH117089 | Mingshan Xue |

The funders had no role in study design, data collection, and interpretation, or the decision to submit the work for publication.

### Author contributions
Dongwon Lee, Data curation, Formal analysis, Investigation, Visualization, Methodology, Writing – original draft, Writing – review and editing; Wu Chen, Data curation, Formal analysis, Supervision, Investigation, Visualization, Methodology, Writing – original draft, Writing – review and editing; Heet Naresh Kaku, Data curation, Software, Formal analysis, Investigation, Visualization, Methodology, Writing – original draft, Writing – review and editing; Xinming Zhuo, Eugene S Chao, Formal analysis, Investigation, Methodology, Writing – review and editing; Armand Soriano, Allen Kuncheria, Stephanie Flores, Joo Hyun Kim, Investigation, Writing – review and editing; Armando Rivera, Investigation, Methodology, Writing – review and editing; Frank Rigo, Supervision, Writing – review and editing; Paymaan Jafar-nejad, Supervision, Methodology, Writing – review and editing; Arthur L Beaudet, Conceptualization, Supervision, Writing – review and editing; Matthew S Caudill, Software, Supervision, Methodology, Writing – review and editing; Mingshan Xue, Conceptualization, Supervision, Funding acquisition, Visualization, Methodology, Writing – original draft, Project administration, Writing – review and editing

### Author ORCIDs
Wu Chen ⓘ https://orcid.org/0000-0002-7400-0519
Armando Rivera ⓘ https://orcid.org/0000-0001-7076-8566
Mingshan Xue ⓘ http://orcid.org/0000-0003-1463-8884

### Ethics
This study was performed in strict accordance with the recommendations in the Guide for the Care and Use of Laboratory Animals of the National Institutes of Health. All of the animals were handled according to approved Institutional Animal Care and Use Committee (IACUC) protocols (AN-6544) of Baylor College of Medicine.

### Decision letter and Author response
Decision letter https://doi.org/10.7554/eLife.81892.sa1
Author response https://doi.org/10.7554/eLife.81892.sa2

---

# Additional files

### Supplementary files
• Supplementary file 1. Comparison of phenotypic rescue of maternal *Ube3a* knockout mice by restoring *Ube3a* expression at different ages. The table summarizes the outcomes of restoring *Ube3a* expression in maternal *Ube3a* knockout mice at different developmental ages from previous studies.

• Supplementary file 2. Primers and probes for reverse transcription droplet digital PCR (RT-ddPCR) and reverse transcription quantitative real-time PCR (RT-qPCR). The sequences of the primers and probes used in the RT-ddPCR or RT-qPCR experiments for detecting *Ube3a*, *Ube3a-ATS*, *Ipw*, and *Gapdh* are provided.

• Supplementary file 3. Statistics of experimental results. The details of all statistical tests, numbers of replicates, and p values are presented for each experiment in the table.

• MDAR checklist

## Data availability

All data generated or analyzed during this study are included in the manuscript and supporting file; Source Data files have been provided for Figure 3, Figure 3-supplement 1, Figure 3-supplement 3, and Figure 3-supplement 4.

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
