## [Editor Report]

We believe this study contributes importantly to the literature and in particular, provides support for the potential value of further postnatal rescue experiments in animal models and perhaps future trials in patients.

---

## [Decision Letter]

**Decision letter after peer review:**

Thank you for submitting your article "Antisense oligonucleotide therapy rescues disturbed brain rhythms and sleep in juvenile and adult mouse models of Angelman syndrome" for consideration by *eLife*. Your article has been reviewed by 3 peer reviewers, and the evaluation has been overseen by a Reviewing Editor and Gary Westbrook as the Senior Editor. The following individual involved in review of your submission has agreed to reveal their identity: Larry Reiter, UT Health Science Center (Reviewer #1). The reviewers have discussed their reviews with one another, and the Reviewing Editor has drafted this to help you prepare a revised submission.

Essential revisions:

Please address these issues fully in your revised submission:

1. Analysis of the particular isoforms of Ube3a deficient mice that are rescued by the ASO treatment is an important component of the analysis and will allow a better understanding of the effects of the ASO treatment in the context of previous studies with developmental rescue. Previous studies have demonstrated the importance of particular isoforms of Ube3a in function. The authors should evaluate the success of the ASO in rescuing expression of these isoforms and discuss these results in the context of the literature.

2. We think you must carefully address the concerns about western blots and potential for duplication of lanes/bands in at least one of the blots. It appears that lanes may have been duplicated and reproduced at different exposures.

3. It is important to address the possibility that this model with ASO rescue may allow analysis of cognitive deficits and their rescue in the mice. This additional information would broaden the significance of the study because it would either indicate that cognitive function may be improved by ASO treatment at postnatal stages, or not. An example would be examining visuospatial behaviors requiring intact hippocampal plasticity and determining if these are rescued.

*Reviewer #1 (Recommendations for the authors):*

The figures, in general, are too crowded and difficult to read. The data in these figures is acceptable, but the format is too busy to actually digest the important points. I suggest they find better ways to show changes in EEG spectra than bar graphs (see below). That said, the findings appear to suggest that at least EEG power and sleep rhythms can be rescued at the juvenile and, to a lesser extent, adult stages. Previous studies have shown that through genetic manipulation, Ube3a can be turned on developmentally in a Ube3a deficient background at embryonic stages with complete rescue of "critical" phenotypes (ataxia, anxiety, repetitive behavior and epilepsy). However, these studies did not examine EEG or sleep disturbance in any detail, and yet claimed that rescue of Ube3a expression after embryonic stages may not have any clinically relevant effects.

I think the authors make the case that at least some clinically relevant phenotypes can be rescued using ASO approaches postnatally in Ube3a deficient mice. Given the potential impact of this manuscript and the novel claim that is somewhat against current dogma in the field, i.e. that rescue after the embryonic stage will have some effect, I would suggest the following experiments:

1. It is critical to know which isoforms are rescued by the ASO treatments in the new Exon6 deletion mouse model. There has been significant emphasis of isoform III in the literature and the nuclear localization of this isoform as being critical to the rescue of predominant phenotypes (Trezza et al. (2019) Nature Neuro 22, pages 1235-1247). The authors need to show rescue of this critical isoform postnatally and/or demonstrate that one of the isoforms is rescued to near normal levels by western blot using tissue from various regions of the brain. Figure 3 – supplemental suggests that some isoforms may be specific to the cerebellum, thalamus, and olfactory bulb and that in the thalamus, at least, there is better rescue of the lower molecular weight protein than the larger isoform. Can you explain this? A better investigation of the particular isoforms being rescued is warranted here.

2. There is a missed opportunity here to test cognition. Did the investigators test these mice for rescue of cognitive phenotypes? If not, why? Hippocampal plasticity is essential for cognition and can be rescued at any age in Ube3a deficient animals (Silva-Santos et al. (2015) J Clin Invest. 125: 2069-76 and others – see Elgersma and Sonzogni (2021) Dev Med and Child Neuro 63: 802-807).

3. Although additional experiments on the impact of other genes in the 15q11.2 critical region (deleted in ~70% of AS individuals) is beyond the scope of the current study, the authors must discuss individual genes, GABA receptors and HERC2, and their impact or involvement in EEG phenotypes in AS. For example, Frolich et al. found that deletion class individuals differ significantly from UBE3A mutation or imprinting center mutation class by spectral power (Frohlich et al. (2019) Biol Psych 85(9): 752-759).

*Reviewer #2 (Recommendations for the authors):*

No effort was made to look at the designed ASO in the original ube3a mouse model. The ability to show good penetration beyond several weeks is an advance in the field and should be built on. However, it is unclear how the newly designed ASO will work in wild type brain or non-neuronal cells. One would expect no change would occur in non-neuronal cells. Many of the rescue experiments appear to be modest but significant differences. The protein rescue appears strong. How about substrates? Is the Ube3a protein functional? Is protein degradation restored? The rescue of the abnormal EEG seems slight and is not different between the control ASO and the ube3a-ASO. Sleep pattern differences look promising and could be expanded upon. The authors report on the finding that poly-spikes are not rescued. It is unclear at this time why this is. It seems that this should be followed up with significant experiments. One possibility is that there is not enough protein being made from the paternal ube3a using the current strategy. The authors could consider repeating their experiments in the original ube3a mouse model to see if similar results are observed. It appears from this study that a best mouse model for AS may not exist. With that in mind, it may be highly beneficial for this group to test several of their ideas on a molecular level and see if in iPSCs derived neurons from AS patients they can observe rescue of cellular and physiological changes.

*Reviewer #3 (Recommendations for the authors):*

This is a well-designed and executed study. The team first developed a new mouse model of Angelman syndrome and then leveraged the model to validate the ASO oligonucleotides therapeutic approach to knock down ATS expression and increase UBE3A expression from the paternal locus. They performed patient-relevant functional assays including EEG rhythms, sleep architecture, and epileptiform activity. The longitudinal design is another major strength of the study, in which both juvenile and adult age groups were included and functional tests were performed at 3-, 6- and 10-weeks post ASO injection. The authors also did a thorough analysis of UBE3A expression levels in different brain regions under different conditions. Overall, the study demonstrated a clear correlation between UBE3A expression level and EEG power ratio (a+b1+b2)/g2 and REM sleep. Improved UBE3A expression was able to restore (to certain degrees) EEG spectra and sleep functions. These results support a cause-and-effect relationship. Interestingly, improved UBE3A expression did not significantly reduce poly-spikes. In addition, the manuscript is well written with a good discussion of the findings in the context of the literature and future perspectives.

I have several concerns.

One concern relates to the Western blot data in Figure 3: there appears to be data duplication and the same blot with different exposures was used for different brain regions. This reviewer believes that it is an honest mistake, so please check each and every blot to ensure the accuracy and fidelity of all data.

Regarding single unilateral ICV injection, there must be variations among animals and it would help if the authors could provide additional data to support the effectiveness and efficiency of the procedure.

Furthermore, the study focused on EEG activity and sleep, and rightly so. If ASO therapy can reduce sleep disturbance, it would likely improve some aspects of cognitive functions. The cognitive improvement would be an important addition to the present work.

---

## [Author Response]

Essential revisions:Please address these issues fully in your revised submission:1. Analysis of the particular isoforms of Ube3a deficient mice that are rescued by the ASO treatment is an important component of the analysis and will allow a better understanding of the effects of the ASO treatment in the context of previous studies with developmental rescue. Previous studies have demonstrated the importance of particular isoforms of Ube3a in function. The authors should evaluate the success of the ASO in rescuing expression of these isoforms and discuss these results in the context of the literature.

We thank the reviewer for this important suggestion, as previous studies showed that the short Ube3a isoform 3 is critical for the pathogenesis of Angelman syndrome. We performed new Western blot experiments to separate the two Ube3a isoforms 2 and 3 that differ by 21 amino acids and about 2-3 kDa. We found that both long and short isoforms were similarly up-regulated by Ube3a-as ASO or Snord115 ASO in different brain regions at 3 weeks post ASO injection into juvenile mice (new Figure 3-supplement 3). We also examined two brain regions from 3 weeks post ASO injection into adult mice and found the same result (new Figure 3-supplement 4). Therefore, these results are consistent with the rescue of EEG power spectrum and sleep phenotypes in maternal *Ube3a* knockout mice.

2. We think you must carefully address the concerns about western blots and potential for duplication of lanes/bands in at least one of the blots. It appears that lanes may have been duplicated and reproduced at different exposures.

We thank the reviewer for pointing out this critical issue and have inspected all raw data and figures. We realized that in the previous Figure 3-supplement 1A, both images of the “Olfactory bulb” and “Midbrain + Hindbrain” anti-Ube3a blots from “3 weeks post ASO injection to juvenile mice” came from the olfactory bulb blot. We have identified the process that led to this error.

The two original image files assigned with the Image ID #179 and #182 were generated by LI-COR Image Studio software on August 14, 2019. Multiple images (#179-1, #179-2, #182-1, #182-2, #182-3) were then exported from the original image files with different exposure settings that were adjusted for individual blots (see Author response image 1). Images #179 have 2 blots (Blot A: anti-Ube3a from Olfactory bulb, Blot B: anti-Gapdh from Olfactory bulb). However, there was dust partially covering the Ube3a band in the second lane of Blot A (see the cyan box in Image #179-2) and its magnification shown as Image #179-2 MAG”. After washing off the dust, this Blot A was re-imaged with two other blots (Blot C: anti-Ube3a from Midbrain + Hindbrain, Blot D: anti-Gapdh from Midbrain + Hindbrain) in Images #182. For Olfactory bulb, Blot A in Image #182-1 and Blot B in Image #179-1 were correctly selected for anti-Ube3a and anti-Gapdh, respectively. For Midbrain + Hindbrain, Blot D in Image #182-3 was correctly selected for anti-Gapdh. Blot C in Image #182-2 (red box) should have been selected for anti-Ube3a, but unfortunately, Blot A in Image #182-2 was mistakenly selected for anti-Ube3a. This is why the image of “Midbrain + Hindbrain” anti-Ube3a blot in the previous Figure 3-supplement 1A is the same as the one from olfactory bulb with a different exposure setting. We have now corrected this mistake in Figure 3-supplement 1A and its source data by using the Blot C in Image #182-2.

**Author response image 1. sa2fig1:** 3 weeks post ASO injection to juvenile mice.

3. It is important to address the possibility that this model with ASO rescue may allow analysis of cognitive deficits and their rescue in the mice. This additional information would broaden the significance of the study because it would either indicate that cognitive function may be improved by ASO treatment at postnatal stages, or not. An example would be examining visuospatial behaviors requiring intact hippocampal plasticity and determining if these are rescued.

We agree with the reviewers that it would have been informative to test the effect of ASO treatment on cognitive function, as it is severely affected in Angelman patients. However, we chose not to test cognitive function in the mouse models of Angelman syndrome for two reasons.

The first reason is that the cognitive deficit is unfortunately a key phenotype that was not well recapitulated in the mouse models, which does not allow a robust test of the ASO therapy. The original maternal *Ube3a* knockout mice (*Ube3a^m∆e5/p+^*) were tested in Morris water maze and fear conditioning, but the deficits are mild and highly variable across labs and strains (for example, see Jiang, et al., 1998; van Woerden et al., 2007; Daily et al., 2011; Huang et al., 2013; Born et al., 2017). This issue has been discussed and summarized by Sonzogni et al., 2018 and Rotaru, et al., 2020.

For Morris water maze, Sonzogni et al., 2018 discussed that “we found that a large number of mice are needed to detect significant differences and results varied strongly among experimenters.” For fear conditioning, Sonzogni et al., 2018 discussed that “we have not been able to get consistent results across experiments and experimenters” and “…Collectively, these studies indicate that this phenotype is rather weak, and hence results, obtained with these tests should be interpreted with care.” The Figure 2 of Rotaru, et al., 2020 summarizes the cognitive phenotypes of Angelman mouse models being “not present/mild”. We believe that most likely for this reason, most of the recent studies testing various genetic therapies in Angelman mouse models did not test cognitive function (see Wolter et al., 2020; Milazzo et al., 2021; Schmid et al., 2021). Judson et al., 2021 did test fear conditioning in the AAV gene replacement experiments, but also stated that “Common learning paradigms in mice – such as Morris water maze and fear conditioning – expose rather mild and inconsistent deficits in the AS model that demand intensive statistical sampling (43). Accordingly, we obtained inconclusive results when fear conditioning adult WT and AS mice treated with PHP.B/hUBE3Aopt or vehicle as neonates (Supplemental Figure 4A). Although mice of each group clearly demonstrated associative learning of the unconditioned foot shock stimulus and conditioned auditory cue (Supplemental Figure 4B), we were underpowered to detect robust deficits in either contextual or cued fear memory in the AS + vehicle group – or rescue thereof (Supplemental Figure 4, C and D).” Note that this experiment used 28, 21, 26, and 26 mice for the 4 groups of mice, yet was still not able to detect the phenotype. It is currently unknown why the cognitive phenotype is mild in the mouse models despite their hippocampal plasticity deficits, but perhaps is due to a compensatory mechanism.

The second reason is that our colleague Dr. Rodney Samaco in our institute has systematically compared the original model (*Ube3a^m∆e5/p+^*) and our new model (*Ube3a^m∆e6/p+^*) in different neurobehaviors including cognitive function tests and found that the two models were very similar. Hence, at the outset of our project, we did not plan to test cognitive function or any other neurobehaviors to avoid overlaps with his work. Our goal was to test the EEG power spectrum and sleep phenotypes, which are clinically highly relevant but not examined in any previous genetic therapy experiments.

Therefore, we hope that the reviewers will agree that testing the effect of ASO treatment on the cognitive function of Angelman mouse models is unlikely to be informative. This would be best tested in a different model that recapitulates the patient phenotype better than the mouse models.

Reviewer #1 (Recommendations for the authors):The figures, in general, are too crowded and difficult to read. The data in these figures is acceptable, but the format is too busy to actually digest the important points. I suggest they find better ways to show changes in EEG spectra than bar graphs (see below). That said, the findings appear to suggest that at least EEG power and sleep rhythms can be rescued at the juvenile and, to a lesser extent, adult stages. Previous studies have shown that through genetic manipulation, Ube3a can be turned on developmentally in a Ube3a deficient background at embryonic stages with complete rescue of "critical" phenotypes (ataxia, anxiety, repetitive behavior and epilepsy). However, these studies did not examine EEG or sleep disturbance in any detail, and yet claimed that rescue of Ube3a expression after embryonic stages may not have any clinically relevant effects.I think the authors make the case that at least some clinically relevant phenotypes can be rescued using ASO approaches postnatally in Ube3a deficient mice. Given the potential impact of this manuscript and the novel claim that is somewhat against current dogma in the field, i.e. that rescue after the embryonic stage will have some effect, I would suggest the following experiments:1. It is critical to know which isoforms are rescued by the ASO treatments in the new Exon6 deletion mouse model. There has been significant emphasis of isoform III in the literature and the nuclear localization of this isoform as being critical to the rescue of predominant phenotypes (Trezza et al. (2019) Nature Neuro 22, pages 1235-1247). The authors need to show rescue of this critical isoform postnatally and/or demonstrate that one of the isoforms is rescued to near normal levels by western blot using tissue from various regions of the brain. Figure 3 – supplemental suggests that some isoforms may be specific to the cerebellum, thalamus, and olfactory bulb and that in the thalamus, at least, there is better rescue of the lower molecular weight protein than the larger isoform. Can you explain this? A better investigation of the particular isoforms being rescued is warranted here.

We thank the reviewer for this important suggestion, as previous studies showed that the short Ube3a isoform 3 is critical for the pathogenesis of Angelman syndrome. We performed new Western blot experiments to separate the two Ube3a isoforms 2 and 3 that differ by 21 amino acids and about 2-3 kDa. We found that both long and short isoforms were similarly up-regulated by Ube3a-as ASO or Snord115 ASO in different brain regions at 3 weeks post ASO injection into juvenile mice (new Figure 3-supplement 3). We also examined two brain regions from 3 weeks post ASO injection into adult mice and found the same result (new Figure 3-supplement 4). Therefore, these results are consistent with the rescue of EEG power spectrum and sleep phenotypes in maternal *Ube3a* knockout mice.

In the original and new Figure 3-supplement 1, the additional bands above the Ube3a bands are nonspecific band because (1) their levels do not differ between WT and maternal *Ube3a* knockout mice and (2) the long and short Ube3a isoforms differ only by 2-3 kDa and were not separated in these blots.

2. There is a missed opportunity here to test cognition. Did the investigators test these mice for rescue of cognitive phenotypes? If not, why? Hippocampal plasticity is essential for cognition and can be rescued at any age in Ube3a deficient animals (Silva-Santos et al. (2015) J Clin Invest. 125: 2069-76 and others – see Elgersma and Sonzogni (2021) Dev Med and Child Neuro 63: 802-807).

The reviewer is correct that hippocampal plasticity can be rescued in adult maternal *Ube3a* knockout mice. However, we did not test our mice for rescue of cognitive phenotypes for two reasons.

The first reason is that the cognitive deficit is unfortunately a key phenotype that was not well recapitulated in the mouse models, which does not allow a robust test of the ASO therapy. The original maternal *Ube3a* knockout mice (*Ube3a^m∆e5/p+^*) were tested in Morris water maze and fear conditioning, but the deficits are mild and highly variable across labs and strains (for example, see Jiang, et al., 1998; van Woerden et al., 2007; Daily et al., 2011; Huang et al., 2013; Born et al., 2017). This issue has been discussed and summarized by Sonzogni et al., 2018 and Rotaru, et al., 2020.

For Morris water maze, Sonzogni et al., 2018 discussed that “we found that a large number of mice are needed to detect significant differences and results varied strongly among experimenters.” For fear conditioning, Sonzogni et al., 2018 discussed that “we have not been able to get consistent results across experiments and experimenters” and “…Collectively, these studies indicate that this phenotype is rather weak, and hence results, obtained with these tests should be interpreted with care.” The Figure 2 of Rotaru, et al., 2020 summarizes the cognitive phenotypes of Angelman mouse models being “not present/mild”. We believe that most likely for this reason, most of the recent studies testing various genetic therapies in Angelman mouse models did not test cognitive function (see Wolter et al., 2020; Milazzo et al., 2021; Schmid et al., 2021). Judson et al., 2021 did test fear conditioning in the AAV gene replacement experiments, but also stated that “Common learning paradigms in mice – such as Morris water maze and fear conditioning – expose rather mild and inconsistent deficits in the AS model that demand intensive statistical sampling (43). Accordingly, we obtained inconclusive results when fear conditioning adult WT and AS mice treated with PHP.B/hUBE3Aopt or vehicle as neonates (Supplemental Figure 4A). Although mice of each group clearly demonstrated associative learning of the unconditioned foot shock stimulus and conditioned auditory cue (Supplemental Figure 4B), we were underpowered to detect robust deficits in either contextual or cued fear memory in the AS + vehicle group – or rescue thereof (Supplemental Figure 4, C and D).” Note that this experiment used 28, 21, 26, and 26 mice for the 4 groups of mice, yet was still not able to detect the phenotype. It is currently unknown why the cognitive phenotype is mild in the mouse models despite their hippocampal plasticity deficits, but perhaps is due to a compensatory mechanism.

The second reason is that our colleague Dr. Rodney Samaco in our institute has systematically compared the original model (*Ube3a^m∆e5/p+^*) and our new model (*Ube3a^m∆e6/p+^*) in different neurobehaviors including cognitive function tests and found that the two models were very similar. Hence, at the outset of our project, we did not plan to test cognitive function or any other neurobehaviors to avoid overlaps with his work. Our goal was to test the EEG power spectrum and sleep phenotypes, which are clinically highly relevant but not examined in any previous genetic therapy experiments.

Therefore, we hope that the reviewer will agree that testing the effect of ASO treatment on the cognitive function of Angelman mouse models is unlikely to be informative. This would be best tested in a different model that recapitulates the patient phenotype better than the mouse models.

3. Although additional experiments on the impact of other genes in the 15q11.2 critical region (deleted in ~70% of AS individuals) is beyond the scope of the current study, the authors must discuss individual genes, GABA receptors and HERC2, and their impact or involvement in EEG phenotypes in AS. For example, Frolich et al. found that deletion class individuals differ significantly from UBE3A mutation or imprinting center mutation class by spectral power (Frohlich et al. (2019) Biol Psych 85(9): 752-759).

We agree with the reviewer that this is an important issue and have now expanded the discussion on ASO therapy in deletion Angelman syndrome.

Reviewer #2 (Recommendations for the authors):No effort was made to look at the designed ASO in the original ube3a mouse model. The ability to show good penetration beyond several weeks is an advance in the field and should be built on. However, it is unclear how the newly designed ASO will work in wild type brain or non-neuronal cells. One would expect no change would occur in non-neuronal cells.

We previously tested Ube3a-as ASO, which is the better one of the two ASOs used in the current study, in the original *Ube3a^m∆e5/p+^* mice (Meng, et al., 2015). The upregulation of paternal Ube3a was similar to what we observed in the new *Ube3a^m∆e6/p+^* mice. Therefore, we did not test the ASOs in the original *Ube3a^m∆e5/p+^* mice again.

We tested both Ube3a-as ASO and Snord115 ASO in WT mice prior to the rescue experiments and found that they increased *Ube3a* mRNA levels. We have now included the results in new Figure 2-supplement 1.

The protein rescue appears strong. How about substrates? Is the Ube3a protein functional? Is protein degradation restored?

Reactivation of paternal *Ube3a* allele produces both Ube3a isoforms with correct sizes (see Milazzo et al., 2021 and our new Figure 3-supplement 3 and Figure 3-supplement 4) and rescues neurological phenotypes. Hence, we believe that the Ube3a protein is functional. We also think that the dysregulated substrates and protein degradation in *Ube3a^m∆e6/p+^* mice should be at least partially restored by Ube3a-as ASO and Snord115 ASO. Although experimentally studying these questions is outside of the scope of our current study, we agree with the reviewer that it is important to test these predictions in the future.

The rescue of the abnormal EEG seems slight and is not different between the control ASO and the ube3a-ASO. Sleep pattern differences look promising and could be expanded upon.

We agree with the reviewer’s assessment and will plan new projects to further study the sleep phenotypes in the future.

The authors report on the finding that poly-spikes are not rescued. It is unclear at this time why this is. It seems that this should be followed up with significant experiments. One possibility is that there is not enough protein being made from the paternal ube3a using the current strategy. The authors could consider repeating their experiments in the original ube3a mouse model to see if similar results are observed. It appears from this study that a best mouse model for AS may not exist. With that in mind, it may be highly beneficial for this group to test several of their ideas on a molecular level and see if in iPSCs derived neurons from AS patients they can observe rescue of cellular and physiological changes.

We agree that it is unclear why the poly-spikes are not rescued by the ASOs. We discussed the two possibilities that suppression of poly-spikes requires up-regulation of Ube3a (1) starting at a younger age and (2) reaching to a higher level than what we have achieved, both of which should be tested in future studies. We also thank the reviewer for other suggestions including testing the original *Ube3a^m∆e5/p+^* mice and iPSC derived neurons, all of which will be helpful for our future studies.

Reviewer #3 (Recommendations for the authors):This is a well-designed and executed study. The team first developed a new mouse model of Angelman syndrome and then leveraged the model to validate the ASO oligonucleotides therapeutic approach to knock down ATS expression and increase UBE3A expression from the paternal locus. They performed patient-relevant functional assays including EEG rhythms, sleep architecture, and epileptiform activity. The longitudinal design is another major strength of the study, in which both juvenile and adult age groups were included and functional tests were performed at 3-, 6- and 10-weeks post ASO injection. The authors also did a thorough analysis of UBE3A expression levels in different brain regions under different conditions. Overall, the study demonstrated a clear correlation between UBE3A expression level and EEG power ratio (a+b1+b2)/g2 and REM sleep. Improved UBE3A expression was able to restore (to certain degrees) EEG spectra and sleep functions. These results support a cause-and-effect relationship. Interestingly, improved UBE3A expression did not significantly reduce poly-spikes. In addition, the manuscript is well written with a good discussion of the findings in the context of the literature and future perspectives.I have several concerns.One concern relates to the Western blot data in Figure 3: there appears to be data duplication and the same blot with different exposures was used for different brain regions. This reviewer believes that it is an honest mistake, so please check each and every blot to ensure the accuracy and fidelity of all data.

We thank the reviewer for pointing out this critical issue and have inspected all raw data and figures. We realized that in Figure 3-supplement 1A, both images of the “Olfactory bulb and “Midbrain + Hindbrain” anti-Ube3a blots from “3 weeks post ASO injection to juvenile mice” came from the olfactory bulb blot. We have identified the process that led to this error.

The two original image files assigned with the Image ID #179 and #182 were generated by LI-COR Image Studio software on August 14, 2019. Multiple images (#179-1, #179-2, #182-1, #182-2, #182-3) were then exported from the original image files with different exposure settings adjusted for individual blots (see Author response image 1). Images #179 have 2 blots (Blot A: anti-Ube3a from Olfactory bulb, Blot B: anti-Gapdh from Olfactory bulb). However, there was dust partially covering the Ube3a band in the second lane of Blot A (see the cyan box in Image #179-2) and its magnification shown as Image #179-2 MAG”. After washing off the dust, this Blot A was re-imaged with two other blots (Blot C: anti-Ube3a from Midbrain + Hindbrain, Blot D: anti-Gapdh from Midbrain + Hindbrain) in Images #182. For Olfactory bulb, Blot A in Image #182-1 and Blot B in Image #179-1 were correctly selected for anti-Ube3a and anti-Gapdh, respectively. For Midbrain + Hindbrain, Blot D in Image #182-3 was correctly selected for anti-Gapdh. Blot C in Image #182-2 (red box) should have been selected for anti-Ube3a, but unfortunately, Blot A in Image #182-2 was mistakenly selected for anti-Ube3a. This is why the image of “Midbrain + Hindbrain” anti-Ube3a blot in the previous Figure 3-supplement 1A is the same as the one from olfactory bulb with a different exposure setting. We have now corrected this mistake in Figure 3-supplement 1A and its source data by using the Blot C in Image #182-2.

Regarding single unilateral ICV injection, there must be variations among animals and it would help if the authors could provide additional data to support the effectiveness and efficiency of the procedure.

The reviewer is correct that for the ICV injection, there are variations among mice, which is probably best reflected by the Western blot results that the Ube3a levels do vary among mice. However, the experimental variation was minimized by a robust injection protocol. We described the method in detail. We also showed that 250 µg ASO/mouse had similar effects as 500 µg ASO/mouse (Figure 3-supplement 5), suggesting that we have likely reached the maximal effect.

Furthermore, the study focused on EEG activity and sleep, and rightly so. If ASO therapy can reduce sleep disturbance, it would likely improve some aspects of cognitive functions. The cognitive improvement would be an important addition to the present work.

We agree with the reviewer that it is of high interest to test the effect of ASO treatment on cognitive function. However, we chose not to test cognitive function in the mouse models of Angelman syndrome for two reasons.

The first reason is that the cognitive deficit is unfortunately a key phenotype that was not well recapitulated in the mouse models, which does not allow a robust test of the ASO therapy. The original maternal *Ube3a* knockout mice (*Ube3a^m∆e5/p+^*) were tested in Morris water maze and fear conditioning, but the deficits are mild and highly variable across labs and strains (for example, see Jiang, et al., 1998; van Woerden et al., 2007; Daily et al., 2011; Huang et al., 2013; Born et al., 2017). This issue has been discussed and summarized by Sonzogni et al., 2018 and Rotaru, et al., 2020.

For Morris water maze, Sonzogni et al., 2018 discussed that “we found that a large number of mice are needed to detect significant differences and results varied strongly among experimenters.” For fear conditioning, Sonzogni et al., 2018 discussed that “we have not been able to get consistent results across experiments and experimenters” and “…Collectively, these studies indicate that this phenotype is rather weak, and hence results, obtained with these tests should be interpreted with care.” The Figure 2 of Rotaru, et al., 2020 summarizes the cognitive phenotypes of Angelman mouse models being “not present/mild”. We believe that most likely for this reason, most of the recent studies testing various genetic therapies in Angelman mouse models did not test cognitive function (see Wolter et al., 2020; Milazzo et al., 2021; Schmid et al., 2021). Judson et al., 2021 did test fear conditioning in the AAV gene replacement experiments, but also stated that “Common learning paradigms in mice – such as Morris water maze and fear conditioning – expose rather mild and inconsistent deficits in the AS model that demand intensive statistical sampling (43). Accordingly, we obtained inconclusive results when fear conditioning adult WT and AS mice treated with PHP.B/hUBE3Aopt or vehicle as neonates (Supplemental Figure 4A). Although mice of each group clearly demonstrated associative learning of the unconditioned foot shock stimulus and conditioned auditory cue (Supplemental Figure 4B), we were underpowered to detect robust deficits in either contextual or cued fear memory in the AS + vehicle group – or rescue thereof (Supplemental Figure 4, C and D).” Note that this experiment used 28, 21, 26, and 26 mice for the 4 groups of mice, yet was still not able to detect the phenotype. It is currently unknown why the cognitive phenotype is mild in the mouse models despite their hippocampal plasticity deficits, but perhaps is due to a compensatory mechanism.

The second reason is that our colleague Dr. Rodney Samaco in our institute has systematically compared the original model (*Ube3a^m∆e5/p+^*) and our new model (*Ube3a^m∆e6/p+^*) in different neurobehaviors including cognitive function tests and found that the two models were very similar. Hence, at the outset of our project, we did not plan to test cognitive function or any other neurobehaviors to avoid overlaps with his work. Our goal was to test the EEG power spectrum and sleep phenotypes, which are clinically highly relevant but not examined in any previous genetic therapy experiments.

Therefore, we hope that the reviewer will agree that testing the effect of ASO treatment on the cognitive function of Angelman mouse models is unlikely to be informative. This would be best tested in a different model that recapitulates the patient phenotype better than the mouse models.